# VideoJudge: Bootstrapping Enables Scalable Supervision of MLLM-as-a-Judge for Video Understanding

**Abdul Waheed   Zhen Wu   Dareen Alharthi   Seungone Kim   Bhiksha Raj**
Carnegie Mellon University
`{abdulw,zhenwu,dalharth,seungonk,bhiksha}@cs.cmu.edu`

 Code    Models & Data

## Abstract

Precisely evaluating video understanding models remains challenging: commonly used metrics such as BLEU, ROUGE, and BERTScore fail to capture the nuances of human judgment, while obtaining such judgments through manual evaluation is costly. Recent work has explored using large language models (LLMs) or multimodal LLMs (MLLMs) as evaluators, but their extension to video understanding remains relatively unexplored. In this work, we introduce VideoJudge, a 3B and 7B-sized MLLM judge specialized to evaluate outputs from video understanding models (*i.e.*, text responses conditioned on videos). To train VideoJudge, our recipe builds on the interplay between a generator and an evaluator: the generator is prompted to produce responses conditioned on a target rating, and responses not matching the evaluator's rating are discarded. Across three out of four meta-evaluation benchmarks, VideoJudge-7B outperforms or is on par with larger MLLM judge baselines such as Qwen2.5-VL (32B and 72B). Notably, we find that LLM judges (Qwen3) models perform worse than MLLM judges (Qwen2.5-VL), and long chain-of-thought reasoning does not improve performance, indicating that providing video inputs is crucial for the evaluation of video understanding tasks.

## 1 Introduction

Recent advances in multimodal large language models (MLLMs) have significantly improved video captioning, question answering, and long-form video understanding across various domains (Nguyen et al., 2024). However, their progress poses a critical challenge: how to evaluate their outputs with reliability, interpretability, and at scale? Traditional reference-based metrics such as BLEU, ROUGE (Lin, 2004), and BERTScore (Zhang et al., 2020) struggle to capture semantic fidelity, contextual grounding, or task-specific reasoning. Moreover, in open-ended tasks where multiple valid answers exist, simple reference overlap can be misleading. Human evaluation is often considered the gold standard, but is expensive, slow to scale, and suffers from inter-annotator variability (Liang et al., 2025). A promising alternative is *LLM-as-a-Judge*. By prompting or fine-tuning language models to assess responses, this paradigm has improved evaluation in text generation (Zheng et al., 2023; Kim et al., 2024a; Gu et al., 2025; Li et al., 2024b) and more recently in vision–language tasks via *MLLM-as-a-Judge* (Chen et al., 2024a; Xiong et al., 2025; Lee et al., 2024b).

Applying MLLM-as-a-judge to video understanding remains underexplored, largely due to the temporal and multimodal complexity of the video. Beyond this inherent difficulty, two broader limitations persist. First, the field lacks large-scale evaluation resources: there are no comprehensive datasets with human preference signals or standardized benchmarks for verifying alignment with human judgments. As a result, existing work either relies on proprietary models such as GPT-4 or GPT-4o (Pu et al., 2025a), which lack transparency and reproducibility, or on small open-source MLLMs in zero-shot settings, which fall short of human-level reliability. Second, principled evaluation criteria are missing. Current (M)LLM-as-a-judge methods depend either on generic rubrics, which are often vague and brittle, or on manually authored rubrics, which cannot scale across tasks.

To address this gap, we introduce a framework to bootstrap data to train video understanding evaluators. The framework has two key pillars. First, it automatically generates training data by producing candidate responses across a 1–5 rating scale, validating them with an evaluator model, and refining cases where predicted ratings diverge from expectations. These bootstrapped examples are then used to train both pointwise and pairwise judge models. Second, the same process enables the construction of new pointwise and pairwise meta-evaluation benchmarks, providing large-scale, high-quality resources for systematic comparison. For evaluation, video–instruction pairs are sourced from datasets distinct from those used in training to avoid distribution overlap. In this way, our approach eliminates the need for costly human annotation while yielding both robust training data and standardized evaluation suites. Beyond these bootstrapped benchmarks, we evaluate our judge models on existing independent, human-annotated benchmarks in both pointwise and pairwise settings, demonstrating that trained judge models learn generalizable evaluation capabilities that align with human judgment rather than merely fitting to the preferences of the generator-evaluator pipeline.

Further, we train MLLM judge models not only to predict ratings with explanations, but also to generate instance-specific rubrics at test time. This enables fine-grained evaluation that is both interpretable and anchored in explicit evaluation guidelines. Experiments in both pointwise and pairwise settings show that our models match or surpass much larger models while correlating more strongly with human ratings and demonstrating higher sample efficiency.

In summary, our contributions are:

- We introduce **VideoJudge**, the first bootstrapped framework for training scalable MLLM-based evaluators across diverse video understanding tasks.
- Train judge models that can not only assign ratings but also generate high-quality, instance-specific rubrics at inference time.
- We demonstrate that fine-tuned small models on the bootstrapped data can match or outperform much larger models in accuracy and alignment with human-specified ratings.
- We provide a suite of trained pointwise and pairwise judge models, meta-evaluation benchmarks, bootstrapped datasets, and other artifacts to support reproducible research in video understanding evaluation.

## 2 RELATED WORKS

**Video Understanding Models and Evaluation** Advances in large language models have led to the development of multimodal models that jointly process and generate outputs across text, image, audio, and video modalities (Bai et al., 2025b; Chen et al., 2024c; Wu et al., 2024b; Xu et al., 2025; Zhao et al., 2025; Chen et al., 2025; Wu et al., 2025). A growing line of work explores video understanding specifically, either by pre-training multimodal models with video–text corpora (Zhang et al., 2024b; 2023; Cheng et al., 2024; Zhang et al., 2025a; Wang et al., 2025) or by instruction-tuning to align video representations with downstream tasks (Zhang et al., 2024c;a). These models are often evaluated using conventional, automatic metrics such as BLEU (Papineni et al., 2002), ROUGE (Lin, 2004), and BERTScore (Zhang et al., 2020), which all assume the existence of reference answers. Human evaluation is also widely used, but it is costly and inconsistent. These limitations motivate more principled automatic and semi-automatic approaches.

**LLM-as-Judge** An alternative paradigm for evaluation uses LLMs themselves as evaluators. Several works have investigated the viability of prompting powerful models such as GPT-4 to act as judges on text generation tasks (Zheng et al., 2023; Liu et al., 2023; Ye et al., 2023; Kim et al., 2025a). Beyond prompting, other efforts fine-tune open-weight models such as Llama-2 (Touvron et al., 2023) and Mistral (Jiang et al., 2023) to serve as reliable evaluators by distilling from GPT-4's assessment trajectories (Kim et al., 2023; 2024b; 2025a). More recently, researchers have extended this line of work to multimodal settings. For example, Chen et al. (2024b) examines whether multimodal LLMs can act as reliable judges, while Lee et al. (2024a) explores fine-tuning open-weight models such as LLaVA-1.5 (Liu et al., 2024) to mimic the evaluation capability of proprietary MLLMs. Similarly, He et al. (2024) and Ku et al. (2024) investigate the use of MLLMs as judges for text-to-image and text-to-video tasks. Together, these works highlight the promise of LLM-as-a-Judge for scalable evaluation, while underscoring the need to further test its robustness in video understanding.

## 3 METHODOLOGY

Our bootstrapping framework consists of a generator–evaluator pipeline that jointly synthesizes data and enforces quality control. The design draws inspiration from self-refinement approaches, where self-consistency (Mitchell et al., 2022; Wang et al., 2023; Chen et al., 2023) and self-verification (Weng et al., 2023) enhance LLM performance, and models adapt through verbal feedback (Madaan et al., 2023). Our overall framework has two stages: (1) iterative bootstrapping to construct large-scale, fine-grained training data, and (2) fine-tuning judge models which are evaluated under both pointwise and pairwise settings. The proposed framework is shown in Figure 1.

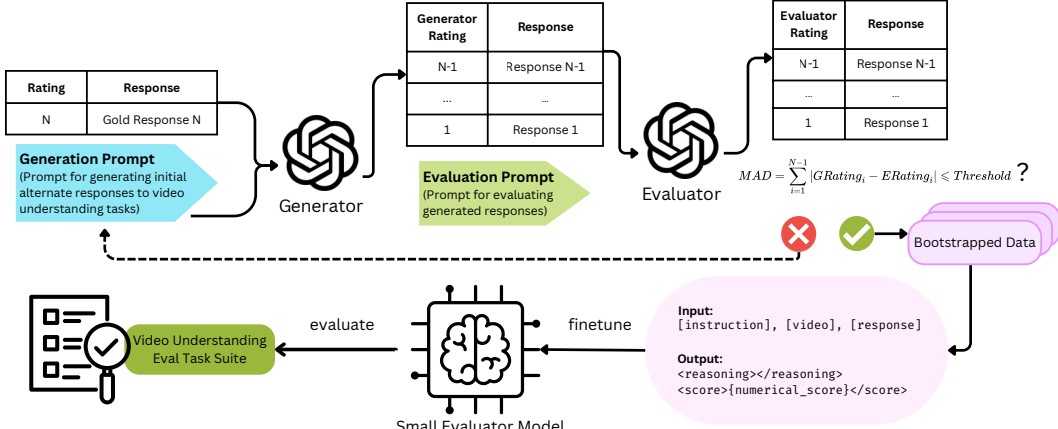

Figure 1: Overview of our bootstrapping framework for training scalable video understanding evaluators. A *generator* first produces candidate responses for a 1 to $N-1$ rating scale ($N = 5$) for each video–instruction pair. These responses are then scored by an *evaluator*, and only candidates whose ratings align with expectations are retained. Through an iterative refinement loop, mismatched responses are revised until they satisfy the acceptance criterion. The resulting bootstrapped dataset provides high-quality supervision signals, which we use to fine-tune small VideoJudge models.

### 3.1 BOOTSTRAPPING PROCESS

We begin with seed data sourced from three large-scale video instruction–response datasets: VideoInstruct-100K (Muhammad Maaz & Khan, 2023), VCG-Plus-112K (Maaz et al., 2024), and VideoChat2-IT (Li et al., 2023). For VideoChat2-IT, which contains multi-turn dialogues, we retain only the first human–assistant exchange. The three corpora are merged and de-duplicated at the instruction level for each video using a MINHASHLSH index (128 permutations, Jaccard threshold 0.9). From this de-duplicated pool, we randomly sample 25K examples, resulting in a corpus of triplets $(v, x, y)$, where $v$ denotes the video, $x$ the instruction, and $y$ the gold-standard response. During bootstrapping, we generate dense video descriptions $\tilde{v}$ using strong vision-language models (§A.2) and use them as semantic context for both the generator and evaluator. This provides richer grounding while reducing the need for repeated inference over raw video, making the pipeline more cost efficient.

To transform this seed corpus into a training dataset for the evaluator, we iteratively generate and refine candidate responses for each $(\tilde{v}, x, y^*)$ triplet. The process follows three stages: *Initial Generation*, *Feedback*, and *Refinement*, described formally below.

**Initial Generation:** For each instruction–video pair $(x, \tilde{v})$ with gold response $y^*$, a generator model $G$ produces $N - 1$ candidate responses, each intended to correspond to a rating $r \in \{1, \ldots, N-1\}$ as shown in 1. The gold response $y^*$ is included as the highest-rated response with rating $N$.

$$y_0^{(r)} = G(p_{\text{gen}}\|\tilde{v}\|x\|y^*, r). \tag{1}$$

**Feedback:** Each candidate response $y_t^{(r)}$ is evaluated by an evaluator model $E$, which assigns a rating $\hat{r}$ and provides reasoning $f_t^{(r)}$. We then compute the deviation between the intended rating

$r$ and the evaluator's rating $\hat{r}$ to determine whether the candidate should be accepted or refined. Candidates for which $\Delta_t^{(r)} \leq \alpha$ are accepted directly into the dataset. The evaluation process can be formalized as:

$$\hat{r}, f_t^{(r)} = E(p_{\text{eval}}\|\tilde{v}\|x\|y^*\|y_t^{(r)}) \qquad (2) \qquad\qquad \Delta_t^{(r)} = |r - \hat{r}| \qquad (3)$$

**Refinement:** For candidates with a rating deviation $\Delta_t^{(r)} > \alpha$, the generator is prompted again using the evaluator's feedback to improve the response. This iterative refinement continues until the candidate meets the acceptance criterion or a maximum of $T$ iterations is reached. The refinement step is formalized as:

$$y_{t+1}^{(r)} = G(p_{\text{ref}}\|\tilde{v}\|x\|y^*\|y_t^{(r)}\|f_t^{(r)}, r) \qquad (4)$$

**Acceptance Criterion:** A candidate response $y_t^{(r)}$ is added to the bootstrapped dataset if $|r - \hat{r}| \leq \alpha$. The final dataset, therefore, consists of $\{(v, x, y, r)\}$ triplets with aligned ratings.

The complete process is outlined in Algorithm 1. Using this pipeline, we bootstrap pointwise data with $N = 5$, where each instruction is paired with five responses rated from 5 to 1. Representative examples are shown in Table 6 in Appendix A.4.

## 3.2 MODEL TRAINING

We use the bootstrapped dataset to train pointwise and pairwise evaluator models. The dataset is defined as $\mathcal{D} = \{(v_i, x_i, y_i, t_i)\}_{i=1}^{M}$, where $v_i$ denotes the video, $x_i$ the instruction, $y_i$ a candidate response (or a response pair in the pairwise setting), and $t_i$ the associated target annotation, such as a rating or a preference label. The evaluator model is trained end-to-end to autoregressively generate the target sequence $t_i$ conditioned on $(v_i, x_i, y_i)$, with the standard negative log-likelihood over tokens:

$$\mathcal{L}(\theta) = -\frac{1}{M} \sum_{i=1}^{M} \sum_{j=1}^{|t_i|} \log P_\theta\big(t_{i,j} \mid t_{i,<j}, v_i, x_i, y_i\big),$$

where $t_{i,j}$ denotes the $j$-th token of $t_i$. This loss is applied to both pointwise and pairwise models. In the pointwise setting, the model produces intermediate reasoning within `<thinking></thinking>` followed by a scalar rating in `<score></score>`, and it can optionally generate task-specific rubrics in `<rubric></rubric>` before reasoning and evaluation. In the pairwise setting, the model outputs its decision within `<answer></answer>` based on a pair of candidate responses.

## 4 EXPERIMENTS

We bootstrap pointwise data starting from 25K seed video instruction-response pairs. After the bootstrapping process, we retain only instructions with at least five responses (one for each rating), resulting in 103,825 examples across 20,765 unique video–instruction pairs.

We construct pairwise supervision by forming response pairs where the higher-rated output is chosen as preferred. Due to computational limitations and while keeping the setting identical, we randomly sample 50% of all possible pairs, resulting in 103,825 pairwise training examples. Both pointwise and pairwise judge models are trained on these bootstrapped datasets. In the *pointwise* setting, the model takes a video, instruction, and candidate response as input, and is trained to produce a reasoning trace followed by a score. We further train a judge model to first generate an instruction-specific evaluation rubric, which is then applied when scoring, ensuring that evaluations are grounded in context-specific criteria. In the *pairwise* setting, each instruction is paired with two candidate responses[1], and the model is trained to identify the preferred response.

---

[1] To avoid positional bias, the order of responses is randomized during both training and evaluation.

## 4.1 BASELINES

We evaluate a broad range of models, including unimodal language models and multimodal video–language models, to evaluate their ability as judges. Unimodal models are tested using detailed video descriptions (§A.2) as proxies for visual input, while video models directly process the video content. This setup allows us to compare how different model classes handle the judging task.

**Unimodal Models** Each unimodal model is provided with the description, instruction, and candidate response, and is prompted to generate a reasoning trace followed by a score. We consider Qwen3 (Yang et al., 2025a) family of models from 0.6B to 14B, and also enable the "thinking mode" of smaller models (up to 4B) to test whether extended reasoning sequences enhance judging ability (Chan et al., 2025; Kim et al., 2025b; Zhou et al., 2025).

**Video Models** We evaluate Qwen2.5-VL(3B–72B) (Bai et al., 2025a) along with other recent video–language models, including LLaVA-Next (7B) (Zhang et al., 2024b), VideoR1 (7B) (Feng et al., 2025), and LLaVA-OneVision (Li et al., 2024a). In our preliminary experiments we find that several models—such as VideoLLaMA3-7B (Zhang et al., 2025b), VideoChat-Flash (Li et al., 2024d), Keye-VL (Yang et al., 2025b), and SmolVLM2 (Marafioti et al., 2025) often failed to follow instructions or produce valid scores under the same evaluation setup. Consequently, we exclude them from our main results.

## 4.2 EVALUATION

**Pointwise** Each video–instruction–response triplet is evaluated independently, with the model producing a reasoning trace followed by a rating on a 1–5 scale. We construct two meta-evaluation benchmarks, VideoJudgeLLaVA-MetaEval and VideoJudgeVCG-MetaEval, by sourcing seed instruction data from LLaVA-Video (Zhang et al., 2024c) and VideoChatGPT [2], then generating additional responses via our bootstrapping pipeline (Algorithm 1) with threshold 0. We report correlation and error metrics, as well as divergence error. We further evaluate on Vatex-Eval (Shi et al., 2022), which contains multiple human judgments aggregated into continuous ground-truth scores, emphasizing ranking- and separation-based measures to capture preference consistency. We also use LongVideoBench (Wu et al., 2024a) for long-form multiple-choice evaluation by rating correct versus distractor answers, reporting both the average score gap (*Delta*) and *pairwise superiority*.

**Pairwise** The judge models compare two candidate responses for the same video–instruction and select the preferred one. We use VideoAutoArena (Luo et al., 2025), where human preferences serve as ground truth. From our pointwise evaluation data, we also construct VideoJudge-Pairwise by pairing responses with different ratings and treating the higher-rated response as correct, measuring accuracy against this derived ground truth. To probe more subtle distinctions, we create VideoJudge-Pairwise-H, focusing on challenging 2-vs.-3 cases: we sample 250 such pairs, collect annotations from two human evaluators, and retain only those with full agreement, yielding over 200 pairs with human preference ground truth. We report the accuracy score for all pairwise evaluations.

**Experimental Setup** All models are trained and evaluated under identical hyperparameter settings to ensure fairness. We perform *full fine-tuning* in *BF16* precision with a maximum sequence length of 128K tokens with fps rate of 1 with maximum number of frames 60 for training and 180 during evaluation. We train all models for 2 epochs with a batch size of 16. The learning rate is set to $2 \times 10^{-7}$ with cosine decay, a warmup ratio of 0.03, weight decay of 0, and gradient clipping at 1. We provide key hyperparameters and other implementation details in Appendix A.7

## 5 DATA EVALUATION

We evaluate the bootstrapped data to ensure the quality and that it provides meaningful and reliable supervision. Our data evaluation has two parts: automatic checks to assess the relative quality of the responses, and human evaluation to validate correctness and preference alignment. Together, these evaluations confirm that the generated data is of sufficient quality for training and benchmarking.

---

[2]`https://huggingface.co/datasets/lmms-lab/VideoChatGPT`

## 5.1 Automatic Evaluation

During our bootstrapping process, we prompt the generator model to produce candidate responses for different ratings by progressively degrading the quality according to the specified score. A natural proxy to verify that the generated dataset adheres to this design is to assess whether response quality indeed declines as we move from higher to lower ratings. To this end, we compute BERTScore and BLEU using the gold response as reference and the generated candidates (ratings 4–1) as hypotheses. The results presented in Figure 2, show a clear monotonic degradation: BERTScore decreases from 91.1 (5–4) to 86.9 (5–1), while BLEU drops from 11.0 to 3.0. This consistent downward

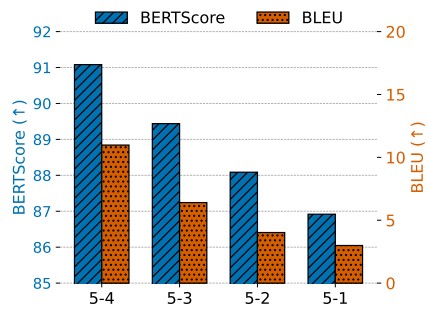

Figure 2: Comparison of **BERTScore** (left axis, blue) and **BLEU** (right axis, orange) across rating pairs. BERTScore remains consistently high while BLEU drops sharply with larger rating gaps.

trend shows that the generator reliably produces responses of progressively lower quality, validating the effectiveness of our controlled response generation process. To further incorporate video context into the evaluation, we also calculate VQAScore (Lin et al., 2024) (§ A.5) with results shown in Figure 16 and observe similar trend.

## 5.2 Human Evaluation

We construct our pairwise data by sampling response pairs with different ratings for the same instruction, choosing the higher-rated one as preferred. In practice, we found that generator–evaluator disagreements were most frequent around ratings 2 and 3, even after incorporating the feedback loop. To focus on these harder cases, we restricted human evaluation to pairs with ratings 2 vs. 3. We sample 250 such examples, each containing the video, a detailed description, the instruction, and two candidate responses. We then ask two annotators to select the preferred response. Results are shown in Table 7. Agreement between annotators is high (94.8% with Cohen's $\kappa$ of 89.5), and both annotators achieved over 92% correctness relative to the gold preference. The preference distribution shows a mild bias toward response b, but error analysis indicates only 4.4% cases where both annotators agreed on the wrong response and 5.2% where they disagreed on a correct one. Overall, human evaluation validates that the generated pairwise data is consistent and reliable, even in the most ambiguous rating regions. We provide detailed metrics of our human evaluation study in Table 7 and examples in Table 8, Appendix A.6.

## 6 Results and Discussion

We train Qwen2.5-VL (3B, 7B) models for both pointwise and pairwise evaluation under identical settings. We evaluate various baselines and our trained judge models across a suite of meta-evaluation benchmarks. We report pointwise evaluation results in Table 1 and pairwise results in Table 3. Our findings show that bootstrapped supervision enables smaller models to reach, and in some cases surpass, the judgment reliability and accuracy of much larger ($\sim 10\times$) general-purpose models. We discuss our findings in subsequent sections.

### 6.1 Pointwise Evaluation

We evaluate all models in the pointwise setup, where each system is required to produce a scalar score. We use the identical prompt decoding parameter across the different models. Unimodal models provide a useful reference point. Across Qwen3 variants, we find that performance on VideoJudgeLLaVA and VideoJudgeVCG is reasonably strong, but VATEX remains challenging with consistently high error and poor calibration. LongVideoBench is considerably more difficult. Although unimodal models achieve non-trivial PSup scores, the gap between correct and distractor responses remains small, suggesting that temporal dependencies are hard to capture from text-only

signals. Thinking mode further improves the 0.6B model, showing that explicit reasoning steps are beneficial. However, enabling unimodal models to perform judgment requires high-quality, detailed descriptions, often generated by powerful models such as Qwen2.5-VL-72B or GPT-4o-mini. As a result, the computational cost of generating these descriptions should be accounted for when considering the overall efficiency of the approach.

Video-language model baselines such as LLaVA-NeXT, OneVision, and Video-R1 perform competitively on VideoJudgeLLaVA and VideoJudgeVCG, achieving correlations in the 0.66–0.77 range with relatively low error values, on par with or better than several unimodal Qwen3 baselines. However, their performance degrades substantially on LongVideoBench, where both PSup and $\Delta$(C–D) drop sharply (e.g., LLaVA-NeXT: 0.59 / 0.45, Video-R1: 0.60 / 0.54), underscoring the difficulty of long-context temporal reasoning. In contrast, the Qwen2.5-VL series scales more robustly: larger variants (32B, 72B) show consistent improvements across all four benchmarks, achieving higher correlations and stronger $\Delta$(C–D) values on LongVideoBench.

Our trained VideoJudge models demonstrate consistently strong performance across most evaluation settings. On VideoJudgeLLaVA and VideoJudgeVCG, both VideoJudge-3B and VideoJudge-7B achieve correlations that not only match but in several cases surpass those of substantially larger baselines such as Qwen2.5-VL-32B/72B, while also outperforming post-trained baseline video-language systems including LLaVA-NeXT, OneVision, and Video-R1. Beyond short-context benchmarks, VATEX shows similar patterns: feedback-guided training reduces error and improves calibration, leading to more accurate and better-grounded predictions. The effect is even clearer on LongVideoBench, where many existing models degrade substantially, while VideoJudge maintains high PSup and $\Delta$(C–D) scores. Together, these findings indicate that feedback supervision supports more consistent and temporally coherent evaluation, particularly for longer and more complex videos. Overall, rubric-supervised judges perform on par with or better than larger video-language models, while remaining comparatively scalable.

Table 1: Benchmark results across VideoJudgeLLaVa, VideoJudgeVCG, VATEX, and LongVidB. Metrics: RMSE/MAE (error), S/P (Spearman/Pearson correlation), ECE (calibration), PSup/ $\Delta$(C-D) (preference).

| Model | | VideoJudgeLLaVa | | | | VideoJudgeVCG | | | | VATEX | | LongVidB | |
|---|---|---|---|---|---|---|---|---|---|---|---|---|---|
| | | RMSE$_\downarrow$ | MAE$_\downarrow$ | S$_\uparrow$ | P$_\uparrow$ | RMSE$_\downarrow$ | MAE$_\downarrow$ | S$_\uparrow$ | P$_\uparrow$ | RMSE$_\downarrow$ | ECE$_\downarrow$ | PSup$_\uparrow$ | $\Delta$(C-D)$_\uparrow$ |
| *Unimodal* | Qwen3-0.6B | 1.20 | 0.92 | 0.64 | 0.63 | 1.20 | 0.92 | 0.66 | 0.65 | 1.44 | 0.84 | 0.58 | 0.17 |
| | Qwen3-1.7B | 0.97 | 0.60 | 0.78 | 0.78 | 1.33 | 0.85 | 0.61 | 0.60 | 2.18 | 0.86 | 0.64 | 0.71 |
| | Qwen3-4B | 0.96 | 0.61 | 0.72 | 0.74 | 1.22 | 0.79 | 0.58 | 0.59 | 2.42 | 0.97 | 0.67 | 0.69 |
| | Qwen3-8B | 0.97 | 0.61 | 0.73 | 0.74 | 1.19 | 0.76 | 0.61 | 0.61 | 2.06 | 0.94 | 0.64 | 0.53 |
| | Qwen3-14B | 1.09 | 0.65 | 0.69 | 0.69 | 1.29 | 0.83 | 0.58 | 0.58 | 2.38 | 0.96 | 0.63 | 0.57 |
| *Unimodal (thinking)* | Qwen3-0.6B | 1.12 | 0.81 | 0.64 | 0.64 | 1.31 | 0.98 | 0.59 | 0.59 | 1.98 | 0.89 | 0.62 | 0.63 |
| | Qwen3-1.7B | 1.00 | 0.70 | 0.73 | 0.75 | 1.34 | 0.94 | 0.56 | 0.58 | 1.99 | 0.82 | 0.65 | 0.86 |
| | Qwen3-4B | 1.02 | 0.69 | 0.68 | 0.70 | 1.33 | 0.90 | 0.51 | 0.52 | 2.12 | 0.92 | 0.65 | 0.69 |
| *Video Models* | LLaVA-NeXT-7B | 1.06 | 0.72 | 0.67 | 0.66 | 1.06 | **0.67** | 0.70 | 0.69 | 1.65 | 0.84 | 0.59 | 0.45 |
| | LLaVA-OneVision | 1.01 | 0.71 | 0.77 | 0.75 | 1.00 | 0.69 | **0.77** | 0.76 | 1.52 | 0.78 | 0.64 | 0.83 |
| | Video-R1-7B | 1.07 | 0.67 | 0.73 | 0.73 | 1.74 | 1.21 | 0.46 | 0.47 | 1.87 | 0.72 | 0.6 | 0.54 |
| | Qwen2.5-VL-3B | 1.31 | 0.94 | 0.63 | 0.63 | 1.58 | 1.12 | 0.51 | 0.52 | 2.27 | 0.85 | 0.56 | 0.20 |
| | Qwen2.5-VL-7B | 0.92 | 0.61 | 0.77 | 0.76 | 1.22 | 0.76 | 0.65 | 0.65 | 2.36 | 0.88 | 0.57 | 0.35 |
| | Qwen2.5-VL-32B | **0.87** | 0.59 | 0.80 | 0.79 | 1.05 | 0.75 | 0.69 | 0.70 | 1.43 | 0.81 | **0.73** | 1.08 |
| | Qwen2.5-VL-72B | **0.87** | 0.61 | 0.80 | 0.81 | **0.98** | 0.69 | 0.76 | **0.77** | 1.40 | 0.79 | 0.71 | 1.06 |
| *Ours* | VideoJudge-3B | 1.07 | 0.61 | **0.82** | **0.82** | 1.59 | 1.06 | 0.59 | 0.63 | **1.33** | **0.63** | 0.61 | 0.70 |
| | VideoJudge-7B | 0.96 | **0.52** | 0.78 | 0.80 | 1.20 | 0.72 | 0.74 | 0.76 | 1.46 | 0.64 | 0.66 | **1.16** |

**Training Judge Models to Generate Instance-Specific Rubrics at Test Time** In our setup, we first synthesize training rubrics and then train the model to (i) generate a rubric for each instance, (ii) reason with the rubric, and (iii) output an integer score. This approach enables scalable, rubric-driven evaluation tailored to individual examples. For computational feasibility, we train *Qwen2.5-VL-3B* on 10% of total pointwise data and evaluate on 1,000 examples sampled from VideoJudgeLLaVA and VideoJudgeVCG. We report the results in Table 2. The rubric generation prompt is shown in Figure 10, and the training/evaluation prompt is provided in Figure 12 in Appendix A.3.

Our results show that *VideoJudgeR-3B*, trained to generate instance-specific rubrics, substantially improves over the 3B and 7B baselines. It reduces error (MAE 0.59 vs. 1.15, RMSE 1.05 vs. 1.56) and achieves correlations above 73, comparable to the much larger 32B and 72B base models. This demonstrates that rubric-driven supervision can close most of the performance gap without scaling model size, yielding evaluations that are both more reliable and more interpretable.

**Evaluating the Quality of Generated Rubrics**
While rubric-driven supervision improves model performance, it is also important to verify whether the rubrics themselves are meaningful and useful for evaluation. High-quality rubrics should specify explicit, context-specific criteria, whereas poor ones risk being vague or generic. To assess rubric quality, we use two methods: LLM-as-Judge, where *GPT-4o-mini* ($temperature = 0$) selects the better rubric between two candidates, and human evaluation, where 300 rubric pairs per model are judged by three annotators, with outcomes aggregated by unanimous (as shown in Figure 3) or majority vote (Figure 17). This dual setup measures alignment with both automated LLM judgments and human preferences.

Table 2: Divergence errors and correlation metrics (P = Pearson, S = Spearman) for zero-shot base models and the VideoJudgeR-3B model trained to generate instance-specific rubrics at test time. All models are prompted to produce rubrics together with reasoning and a score.

| Model | MAE$_\downarrow$ | RMSE$_\downarrow$ | P$_\uparrow$ | S$_\uparrow$ |
|---|---|---|---|---|
| Qwen2.5-VL-3B | 1.15 | 1.56 | 37.85 | 37.96 |
| Qwen2.5-VL-7B | 0.86 | 1.22 | 57.09 | 57.26 |
| Qwen2.5-VL-32B | 0.59 | 0.86 | 78.59 | 80.21 |
| Qwen2.5-VL-72B | 0.54 | 0.87 | 78.10 | 78.61 |
| VideoJudgeR-3B | 0.59 | 1.05 | 73.96 | 74.16 |

Our results show that *VideoJudgeR-3B* produces substantially higher-quality rubrics than the 3B, 7B, and 32B baselines across all settings, with large margins under both unanimous and majority human judgments and even stronger gains in LLM-as-Judge evaluation. Against stronger models, it remains competitive: in the LLM-as-Judge setup, VideoJudgeR-3B achieves a 92.7% win rate against GPT-4o-mini and 71.3% against Qwen-72B, consistently maintaining above 50% win rate across all evaluation settings. These findings demonstrate that instance-specific rubric generation enables a compact 3B model to outperform much larger models while producing rubrics preferred by

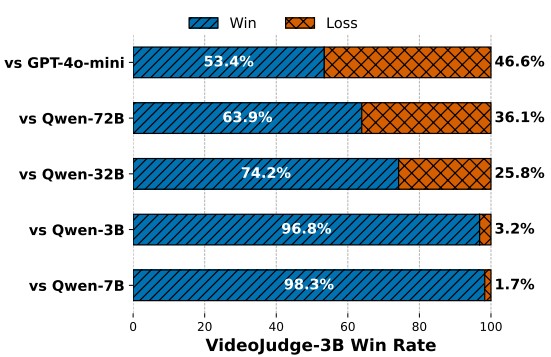

Figure 3: Win rates from human evaluations comparing *VideoJudge-3B* against other models.

both humans and strong LLM judges. We provide example rubrics generated by different models in Table 5 in Appendix A.3.2.

## 6.2 PAIRWISE EVALUATION

We next train and evaluate models in the pairwise setting, where the task is to prefer the better of two responses to the same video–instruction pair. This setup directly captures relative quality and aligns closely with human preference judgments. As before, we assess both base models and our trained VideoJudge models, with and without feedback, across VideoAutoArena (VAA), VideoJudge (VJ), and VideoJudge-Human (VJ-H). Results are summarized in Table 3. VideoJudge models consistently outperform their backbone

Table 3: Accuracy scores ($\uparrow$) of zero-shot base models and VideoJudge on pairwise meta-evaluation benchmarks. Abbreviations: VAA = VideoAutoArena, VJ = VideoJudge, VJ-H = VideoJudge-Human, w/ FB = with feedback, w/o FB = without feedback.

| Model | VAA | | VJ | | VJ-H | |
|---|---|---|---|---|---|---|
| | w/ FB | w/o FB | w/ FB | w/o FB | w/ FB | w/o FB |
| Qwen2.5-VL-3B | 54.90 | 52.16 | 82.60 | 75.00 | 85.23 | 81.01 |
| Qwen2.5-VL-7B | 75.29 | 71.37 | 89.00 | 84.60 | 89.03 | 82.28 |
| Qwen2.5-VL-32B | 80.78 | **90.59** | 91.20 | 91.20 | 92.83 | 90.72 |
| Qwen2.5-VL-72B | **89.80** | _89.80_ | 94.00 | 93.20 | **94.51** | **93.25** |
| VideoJudge-3B | 71.76 | 64.71 | _94.00_ | _95.80_ | 89.45 | 90.72 |
| VideoJudge-7B | _85.49_ | 87.45 | **95.60** | **98.60** | _93.67_ | **93.25** |

baselines across all benchmarks. Notably, **VideoJudge-3B** achieves 94.0 on VJ and 89.45 on VJ-H

(w/ feedback), far surpassing Qwen2.5-VL-3B (82.6 / 85.23) and even outperforming much larger models such as Qwen2.5-VL-32B and 72B in several cases. **VideoJudge-7B** further improves performance, achieving 98.6 on VJ and 93.67 on VJ-H. These results highlight that our bootstrapping process enables smaller models to match or exceed the reliability of much larger video-language systems. Feedback consistently improves the 3B and 7B baselines. For larger models, however, the gains are smaller. In the VideoJudge variants, the effect is more mixed and depends on the benchmark.

**How Many Frames Are Enough for an Effective Video Judge** We study the effect of `maxframes` on judgment performance, as it controls the temporal context available for evaluation. Too few frames risk omitting critical context, while excessively large values increase computation without proportional benefit. To analyze this tradeoff, we vary `maxframes` during training (30–500, evaluation fixed at 180; Figure 20a) and separately during evaluation (30–180, training fixed at 60; Figure 20b). This design isolates the role of temporal coverage in both training and inference.

When varied during training, VideoJudge shows consistent gains from larger `maxframes`. Correlations with ground truth rating increase steadily up to ∼240 frames (exceeding 0.7), while RMSE and MAE decline. Beyond this point, improvements plateau, suggesting diminishing returns despite higher cost. In evaluation, increasing `maxframes` at inference improves correlation and reduces error up to ∼120 frames, after which performance saturates.

Overall, these results indicate that moderate to large temporal context is crucial for effective judgment. Training benefits from up to 240 frames, while at evaluation, modest values (around 120) suffice to capture most relevant evidence. Thus, carefully chosen `maxframes` can balance accuracy and efficiency, strengthening temporal grounding without unnecessary cost.

**Decoding Temperature** We study the effect of decoding temperature on sampling reliability, as it directly controls the trade-off between determinism and diversity. This is particularly important for evaluation models, where stochastic decoding can lead to inconsistent judgments.

Figure 4 (other metrics in Figure 19 in Appendix A.8) reports the pointwise performance of the base Qwen2.5-VL-3B and its VideoJudge-trained counterpart across different temperatures. The base model degrades steadily as temperature increases, with Spearman correlation dropping from 0.56 at $T = 0.0$ to 0.42 at $T = 1.0$, alongside higher error rates and more invalid outputs. In contrast, the VideoJudge model remains stable and even improves at moderate to high temperatures, achieving a peak correlation of 0.73 and the lowest MAE of 0.69.

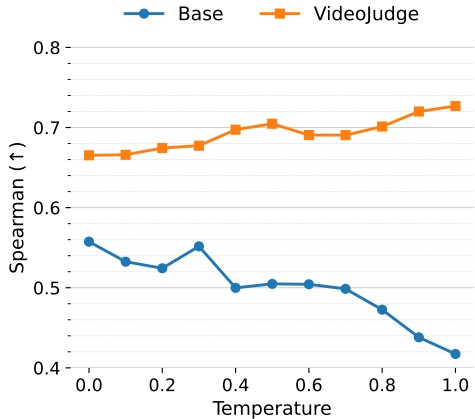

Figure 4: Spearman correlation across temperatures for base and VideoJudge models.

Overall, these results suggest that stochastic decoding introduces variability in judge model outputs, but rubric-guided training significantly improves robustness to temperature changes. This robustness is practically important, as non-deterministic decoding is often used to encourage diverse and well-calibrated evaluations.

**Error Analysis** To better understand where VideoJudge fails, we analyze model evaluations and find a consistent overestimation bias. For example, our judge models overestimate the score by ≥2 points in about 14.8% of cases but underestimate by the same margin in only 1.5%. Calibration in the mid-to-high rating range is also poor: only 36.9% of rating-3 responses get the correct score, with 46.6% inflated to a perfect 5, and 81.3% of rating-4 responses are incorrectly rated as 5. Together, these findings point to the need for training data that emphasizes hard negatives and finer-grained distinctions near the top of the rating scale.

## 7 LIMITATIONS AND FUTURE WORK

While VideoJudge demonstrates strong performance across multiple benchmarks, both the training supervision and a portion of the meta-evaluation benchmarks are constructed using the same generator–evaluator pipeline, which can introduce partial "closed-loop" effects. Although we evaluate, to the best of our knowledge, on the available human-annotated benchmarks, fully independent large-scale pointwise meta-evaluation datasets for video understanding remain limited, constraining evaluation to a relatively small set of benchmarks. Expanding evaluation to larger and more diverse human-annotated datasets would further strengthen the reliability and generalizability of our conclusions. As detailed in our error analysis (§6.2), VideoJudge also exhibits a consistent overestimation bias and poor calibration in the mid-to-high rating range.

Looking ahead, incorporating harder negatives and finer-grained supervision could directly address these calibration gaps, while broader coverage of specialized tasks and domain shifts would strengthen generalization. More broadly, the bootstrapping methodology introduced here is not specific to video understanding and could naturally extend to other multimodal settings where human-annotated evaluation data is scarce.

## 8 CONCLUSION

We introduce VideoJudge, a bootstrapping framework for training MLLM-based evaluators specialized for video understanding. Our approach addresses the lack of evaluation resources with human preference signals and principled evaluation criteria for video understanding. The core contribution lies in an iterative generator-evaluator pipeline that synthesizes training data and enforces quality control, creating over 100,000 training examples without costly human annotation. We then fine-tune judge models to generate both ratings and instance-specific rubrics at test time, enabling interpretable evaluations anchored in explicit criteria grounded in the specific instruction and video content. Our experiments demonstrate that fine-tuned 3B and 7B VideoJudge models match or outperform much larger baselines in accuracy and alignment with human ratings. VideoJudge-3B achieves comparable performance to models up to 10× larger, while VideoJudge-7B consistently outperforms larger video-language models across multiple benchmarks. VideoJudgeR-3B produces rubrics preferred by both human annotators and LLM judges while maintaining performance comparable to much larger base models. By releasing curated meta-evaluation benchmarks, bootstrapped datasets, and trained models, we provide essential resources for reproducible multimodal evaluation research. The bootstrapping methodology is general and could extend to other modalities beyond video understanding.

## 9 ACKNOWLEDGEMENT

We would like to express our sincere thanks to Shaily Bhatt and Kashu Yamazaki for their valuable contributions during the early stages of this project, whose discussions and inputs helped shape its initial direction. We also thank Sagnik Mukherjee and Arif Ahmad for their helpful feedback on the manuscript, which improved the clarity, presentation, and overall quality of the write-up.

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

# A APPENDIX

## A.1 RELATED WORK

**Evaluation Benchmarks**  Video understanding models are evaluated on a wide range of benchmarks spanning different tasks (Sanders & Van Durme, 2024). For captioning, datasets such as MSR-VTT (Xu et al., 2016), VATEX (Wang et al., 2020), and HowTo100M (Miech et al., 2019) provide large-scale paired video–text data, with evaluation often relying on reference-based metrics or correlation with human judgments (Shi et al., 2022). For action recognition, datasets like ActivityNet offer a large-scale benchmark covering hundreds of activity categories with temporal annotations (Caba Heilbron et al., 2015). For video question answering, datasets such as TVQA (Lei et al., 2018) and NEXT-QA (Xiao et al., 2021) require models to integrate visual content with natural language reasoning over semantically complex or temporally extended video segments. In parallel, meta-evaluation benchmarks have been proposed to measure the reliability of evaluators themselves, both in unimodal and multimodal settings. Examples include RewardBench (Lambert et al., 2024) and MM-EVAL (Son et al., 2024) in the unimodal domain, and multimodal resources such as VATEX EVAL (Shi et al., 2022), VLRewardBench (Li et al., 2024c), Judge Anything (Pu et al., 2025b), and LLaVA-Critic (Xiong et al., 2025).

## A.2 DATASET

Table 4: Video duration statistics (in seconds) across evaluation datasets, sorted by number of unique videos. Count indicates unique videos considered after deduplication.

| Duration/Dataset | Count | Min | Max | Mean | Median | Remark |
|---|---|---|---|---|---|---|
| VideoJudge-RS-20K | 9469 | 2.1 | 745.4 | 117.6 | 114.4 | 20K-scale VideoJudge preference dataset |
| VideoJudgeLLaVA-MetaEval | 3038 | 5.0 | 341.8 | 18.3 | 9.9 | VideoJudge benchmark with LLaVA prompts |
| VatexEval-MetaEval | 2340 | 1.9 | 7180.0 | 167.5 | 81.4 | VATEX evaluation split (multilingual captions) |
| VideoJudgeVCG-MetaEval | 499 | 3.0 | 238.0 | 108.0 | 98.7 | VideoJudge benchmark with VCG prompts |
| LongVideoBench-MetaEval | 280 | 8.0 | 297.1 | 56.0 | 16.8 | Long-context video reasoning benchmark |
| VideoAutoArena-Preference | 241 | 8.0 | 3291.4 | 433.6 | 60.7 | Preference pairs from VideoAutoArena |

We provide details of the videos along with the nature and source of instruction data in Table 4. The datasets span a wide range of video durations, from short-clip benchmarks to long-form videos extending to several minutes, reflecting the diversity of evaluation settings covered in our work.

**Video Description Generation**  We generate dense video descriptions using two vision–language models: Qwen2.5-VL-32B-Instruct and GPT-4o-mini (frames are provided as a sequence of images) to use as proxy during bootstraping process. We sample 2 frames per second (FPS) with maximum of 180 frames for each video. We use temperature of 0.6, with all other generation parameters kept at their default settings. We initially use GPT-4o-mini for description generation due to its strong multimodal description quality; however, to scale annotation under practical budget constraints, the remaining descriptions are generated using Qwen2.5-VL-32B-Instruct as a cost-effective alternative. Overall, approximately 33% of the descriptions are generated using GPT-4o-mini and 67% by Qwen2.5-VL-32B-Instruct. We provide the prompt for generating dense video description in Figure 5. For unimodal evaluation, we use Qwen2.5-VL-72B-Instruct for LongVideoBench and VatexEval, and GPT-4o-mini for the remaining benchmarks to balance quality, cost, and efficiency.

### A.3 PROMPTS

In this section, we provide a comprehensive list of prompts that we use in our study.

### A.3.1 BOOTSTRAPPING

**Dense Video Description Generation**    We provide the prompt to generate dense video description in Figure 5.

---

**Video Description Generation Prompt**

You are an assistant specialized in generating **concise, factual video descriptions** based solely on provided **visual information**. When given the content, produce a **coherent and continuous description** in a natural, human-generated paragraph style that accurately captures all **visible elements**, **settings**, and **actions**. Do not refer to the fact that the information comes from video frames, and avoid using bullet points or numbered lists. Do not introduce any additional details, assumptions, or interpretations beyond what is explicitly shown in the input. Prioritize **clarity**, **precision**, and **comprehensive coverage** of the visual information.

**Input:**
Using the video content provided, generate a **detailed and factual video description** that strictly reflects the **visible content**.

---

Figure 5: Prompt used for generating factual video descriptions from visual content. The prompt is used with both GPT-4o-mini and Qwen2.5-VL-32B-Instruct, with frames provided as a sequence of images at 2 FPS and a maximum of 180 frames per video.

**Response Generation Prompt**    The prompt used to generate candidate responses is provided in Figure 6.

**Response Evaluation Prompt**    The prompt for evaluating candidate responses during the bootstrapping stage is shown in Figure 6.

**Response Regeneration from Feedback**    After the initial round of response generation and evaluation, we measure the difference between the generator's self-assigned rating and the evaluator's rating. This numerical gap is then incorporated into feedback, which guides response regeneration. The corresponding prompt is shown in Figure 8.

### A.3.2 TRAINING AND EVALUATION

**Pointwise**    For pointwise evaluation, we prompt the model to produce a reasoning sequence followed by a scalar score, as illustrated in Figure 11.

**Rubric Generation**    We employ *GPT-4o-mini* to construct evaluation rubrics conditioned on the instruction, the video description (serving as a proxy for the video content), and the gold standard response.

We provide example rubrics generated by different models in Table 5

**Pointwise Training, Evaluation, and Rubric Generation**    We use the prompt in Figure 11 to train VideoJudge models and to evaluate responses in a pointwise setup. The prompt in Figure 12 extends this by enabling models to both train and evaluate in a pointwise setting while also generating rubrics at test time. Finally, Figure 13 shows the prompt used to evaluate rubrics produced by different models.

**Pairwise training and evaluation.**    We provide the prompt to train and evaluate models in a pairwise setup without feedback in Figure 14 and with feedback generation in Figure 15.

---

**Candidate Response Generation Prompt**

You are provided with a **detailed video description**, a gold standard **response** rated 5 (perfectly accurate, highest quality), and a corresponding **instruction** for a **video understanding task**. This task may include **video captioning**, **video question answering**, **video instruction following**, **temporal action localization**, or any other open-ended **video understanding** scenario.

Your task is to generate four additional **responses** that simulate progressively lower-quality outputs for the same **instruction**. Each generated **response** should correspond to a quality rating from **4** to **1**, where **Rating 5** is the provided gold standard and **Ratings 4 through 1** represent decreasing quality.

As the rating decreases, the **responses** should reflect increasing levels of degradation, including **hallucinations**, **omissions**, **irrelevant information**, **logical inconsistencies**, or **grammatical issues**. All generated **responses** must remain similar in length to the gold standard and maintain the expected task format (e.g., caption, answer, instruction). Do **not** simply truncate the gold response — simulate **realistic and meaningful degradation** in quality across levels. Use the **video description** to ground the correctness of the response content.

**Task:**
Generate four degraded **responses** corresponding to quality ratings **4** through **1**, based on the gold standard **response** (Rating 5). Follow the rating guidelines strictly.

**Rating Guidelines:**
- **Rating 4**: Mostly accurate with minor issues. Preserves the key meaning with small lapses in detail or precision.
- **Rating 3**: Partially correct. Conveys the general idea but includes noticeable errors, omissions, or misinterpretations.
- **Rating 2**: Poor alignment. Contains serious flaws and only loosely relates to the instruction or video.
- **Rating 1**: Unrelated or incorrect. Fails to reflect the video or follow the instruction meaningfully.

**Example:**
**Instruction:** "Can you give a brief summary of the video content?"
**Video Description:** "A large group of people are participating in an aerobics class inside a spacious indoor venue..."
**Gold Standard Response (Rating 5):** "The video is about a group of people doing a step exercise class, with some confusion among the participants. The focus is on the main woman, who stops at the end and walks towards the camera."
**Rating 4:** "The video shows a group of people in a step exercise class, with some participants appearing confused during the workout. The main woman stops exercising at the end and moves toward the camera."
**Rating 3:** "The video features people doing an aerobics class with stepping movements and some coordination issues. There's a woman who seems to be leading the group, and she approaches the camera at some point during the session."
**Rating 2:** "The video shows several people in what appears to be a dance or fitness routine taking place indoors. A woman in the group walks around and comes closer to where the video is being filmed, while others continue moving in the background."
**Rating 1:** "The video depicts a cooking demonstration where a chef wearing workout clothes explains different healthy recipes to a small audience in what looks like a kitchen or dining area, with people standing around tables."

**Output Format:**
Your final output **must** be a valid JSON object with exactly the following keys: `"rating_4"`, `"rating_3"`, `"rating_2"`, and `"rating_1"`. Each value should be the generated **response** corresponding to that quality rating. Do **not** include any additional commentary, formatting, or explanation outside the JSON object.

**Input:**
**Instruction:**

`{instruction}`

**Video Description:**

`{video_description}`

**Gold Standard Response (Rating 5):**

`{gold_standard_response}`

**Output (according to the JSON schema):**

```
{
  "rating_4": "[Generated response]",
  "rating_3": "[Generated response]",
  "rating_2": "[Generated response]",
  "rating_1": "[Generated response]"
}
```

Figure 6: LLM-as-judge prompt for generating degraded responses at different quality levels based on a gold standard.

---

**Candidate Response Evaluation Prompt**

You are provided with a detailed **video description**, an **instruction**, a gold standard **response** rated 5 (perfectly accurate, highest quality), and a set of candidate **responses** for a **video understanding task**. This task may include **video captioning**, **question answering**, **instruction following**, **temporal action localization**, or any other open-ended **video understanding** scenario.

**Task:**
Your task is to **evaluate each candidate response** and assign an **evaluation rating** from **1** to **4**. While the gold standard **response** can serve as a reference for what an ideal **response** looks like, your evaluation should primarily focus on how well each candidate **response** fulfills the task defined by the **instruction**, given the provided **video description**. Focus your reasoning on identifying what is **incorrect**, **missing**, or **misleading** in the **response** itself.

**Evaluation Steps:**

1. Read the **instruction** carefully to understand the intended task.

2. Refer to the gold standard **response** (rated 5) as a reference for correctness and completeness.

3. Use the provided **video description** as the factual basis for evaluating all responses.

4. For each candidate **response**: identify **inaccuracies**, **omissions**, **hallucinations**, or **irrelevant content**, and evaluate alignment with the **instruction** and **video description**.

5. Assign an integer score from **1–4** to each candidate **response**.

6. Provide reasoning for each rating, focusing on specific aspects that affect alignment with the **instruction** and **video description**.

**Rating Guidelines for Evaluation:**
- **Rating 4**: Mostly accurate with minor issues. Preserves key meaning with small lapses.
- **Rating 3**: Partially correct. General idea conveyed but with noticeable errors/omissions.
- **Rating 2**: Poor alignment. Serious flaws, loosely related to task or video.
- **Rating 1**: Unrelated or incorrect. Fails to reflect the video or follow the instruction.

**Example:**
**Instruction:** "Can you give a brief summary of the video content?"
**Video Description:** "A large group of people are participating in an aerobics class inside a spacious indoor venue..."
**Gold Standard Response (Rating 5):** "The video is about a group of people doing a step exercise class, with some confusion among the participants. The focus is on the main woman, who stops at the end and walks towards the camera."
**Rating 4:** "The video shows a group of people in a step exercise class, with some participants appearing confused during the workout. The main woman stops exercising at the end and moves toward the camera."
**Rating 3:** "The video features people doing an aerobics class with stepping movements and some coordination issues. There's a woman who seems to be leading the group, and she approaches the camera at some point during the session."
**Rating 2:** "The video shows several people in what appears to be a dance or fitness routine taking place indoors. A woman in the group walks around and comes closer to where the video is being filmed, while others continue moving in the background."
**Rating 1:** "The video depicts a cooking demonstration where a chef wearing workout clothes explains different healthy recipes to a small audience in what looks like a kitchen or dining area, with people standing around tables."

**Output Format:**
Your final output **must** follow the exact JSON schema provided. Do **not** include any additional formatting, comments, or text outside of the JSON object.

**Input:**
**Instruction:**

```
{instruction}
```

**Video Description:**

```
{video_description}
```

**Gold Standard Response (Rating 5):**

```
{gold_standard_response}
```

**Generated Responses:**

```
{generated_responses}
```

**Output (according to the JSON schema):**

```
{output_format}
```

Figure 7: LLM-as-judge prompt for evaluating candidate responses against a gold standard using a 1–4 quality scale.

## A.4    BOOSTRAPPING PROCESS

## A.5    EVALUATION DATA

Representative examples of the generated data are shown in Table 6. Figure **??** further illustrates the evaluation results, presenting the BERTScore with the gold-standard response as reference along-

---

**Candidate Response Regeneration with Feedback Prompt**

You are provided with a detailed **video description**, an **instruction**, a gold standard **response** for a **video understanding task** (such as **captioning**, **question answering**, **instruction following**, or **event description**), and a set of generated **responses**, each intended to match a specific **quality rating** from **4** to **1**. However, some generated **responses** were evaluated and found to deviate from their intended quality levels.

**Task:**
Your task is to regenerate revised **responses** only for those entries where the absolute difference between the **intended rating** and the **evaluation rating** is greater than zero (i.e., $|\text{intended\_rating} - \text{eval\_rating}| > 0$). For each such entry, produce a revised **response** that strictly conforms to the **intended quality rating**, based on the definitions in the **rating guidelines** below. Use the **video description** as the factual grounding for determining what content is valid. Adjust the **response** so that its **evaluation rating** would now exactly match the **intended rating**.

- If the **eval_rating** is higher than the **intended_rating**, degrade the **response** by introducing errors such as **hallucinations**, **factual distortions**, **vagueness**, or **grammar issues**.
- If the **eval_rating** is lower than the **intended_rating**, improve the **response** by clarifying actions, reducing errors, or restoring key context from the **video description**.

**Rating Guidelines:**
- **Rating 4**: Mostly accurate with minor issues. Preserves key meaning with small lapses.
- **Rating 3**: Partially correct. Conveys the general idea but with noticeable errors/omissions.
- **Rating 2**: Poor alignment. Serious flaws, loosely related to task or video.
- **Rating 1**: Unrelated or incorrect. Fails to reflect the video or follow the instruction.

**Example:**
**Instruction:** "Can you give a brief summary of the video content?"
**Video Description:** "A large group of people are participating in an aerobics class inside a spacious indoor venue..."
**Gold Standard Response (Rating 5):** "The video is about a group of people doing a step exercise class, with some confusion among the participants. The focus is on the main woman, who stops at the end and walks towards the camera."
**Rating 4:** "The video shows a group of people in a step exercise class, with some participants appearing confused during the workout. The main woman stops exercising at the end and moves toward the camera."
**Rating 3:** "The video features people doing an aerobics class with stepping movements and some coordination issues. There's a woman who seems to be leading the group, and she approaches the camera at some point during the session."
**Rating 2:** "The video shows several people in what appears to be a dance or fitness routine taking place indoors. A woman in the group walks around and comes closer to where the video is being filmed, while others continue moving in the background."
**Rating 1:** "The video depicts a cooking demonstration where a chef wearing workout clothes explains different healthy recipes to a small audience in what looks like a kitchen or dining area, with people standing around tables."

**Output Format:**
Your final output must be a valid **JSON object**. For each entry where $|\text{intended\_rating} - \text{eval\_rating}| > 0$, include a key `"rating_{i}"` and update the value with a newly revised **response** that adheres precisely to the **intended quality rating**. If the **eval_rating** is equal to the **intended_rating**, do not modify or include that entry. Only output the revised **responses**.

**Input:**
**Instruction:**
`{instruction}`

**Video Description:**
`{video_description}`

**Gold Standard Response (Rating 5):**
`{gold_standard_response}`

**Feedback Data (JSON):**
`{feedback_data}`

**Output (according to the JSON schema):**

`{output_format}`

Figure 8: LLM-as-judge prompt for regenerating responses to align with intended quality ratings.

side the absolute VQAScore across different ratings. For BERTScore, we use the *evaluate*[3] library with its default configuration, which uses *roberta-large*[4] model. The VQAScore is computed using

---

[3] https://github.com/huggingface/evaluate
[4] https://huggingface.co/FacebookAI/roberta-large

---

**Evaluating Model-Generated Responses Prompt**

You are provided with a **video**, a corresponding **instruction**, and a **response** generated by a model. The instruction defines a **video understanding task**, which may take any form — including but not limited to **open-ended question answering**, **captioning**, **instruction following**, **temporal reasoning**, or **multi-step inference grounded in the video**. These tasks are open-ended and often require complex or nuanced reasoning over the visual and temporal content.

Your task is to **evaluate the quality of the response**, considering how well it satisfies the task defined by the instruction, based on the content of the video. This is a holistic judgment and should be based on the overall **correctness**, **relevance**, **completeness**, and **grounding** of the response.

**Task:**
For each response:
- Assess how well it addresses the task in the instruction in the context of the video.
- Consider whether the response is accurate, relevant, complete, and grounded in the video.
- Provide a brief rationale explaining the overall quality and alignment of the response with the instruction inside `<thinking> </thinking>`.
- Output a score from 1 (worst) to 5 (best) indicating the overall quality inside `<score> </score>`.

**Rating Guidelines:**
- **Rating 5**: Fully accurate, complete, and well-grounded. The response precisely follows the instruction with no notable errors or omissions.
- **Rating 4**: Mostly accurate with minor issues. The response preserves the key meaning and relevance, with only small lapses in detail or precision.
- **Rating 3**: Partially correct. The response conveys the general idea but includes noticeable errors, omissions, or misinterpretations.
- **Rating 2**: Poor alignment. The response has serious flaws and only loosely relates to the instruction or video.
- **Rating 1**: Unrelated or incorrect. The response fails to reflect the video or follow the instruction meaningfully.

**Input:**
**Instruction:**
`{instruction}`
**Response:**
`{response}`

**Output (Strict Format):**

```
<thinking>{{your reasoning and explanation for the rating}}</thinking>
<score>{{integer score from 1 to 5}}</score>
```

Figure 9: LLM-as-judge prompt for evaluating model-generated responses to video understanding tasks.

*T2V Metrics*[5] with the video model *Qwen2.5-VL-7B-Instruct*[6]. For BLEU, we use the *sacrebleu*[7] implementation with default settings.

## A.6 HUMAN EVALUATION

### A.6.1 PAIRWISE

## A.7 HYPERPARAMETERS

In Table 9 we provide a detailed list of hyperparameters we use in our experiments. We use Qwen2.5-VL [8] as training framework vLLM (Kwon et al., 2023) for evaluation. We keep all other parameters as default until stated otherwise.

## A.8 RESULTS

**Rubric Generation**

---

[5]`https://github.com/linzhiqiu/t2v_metrics`
[6]`https://huggingface.co/Qwen/Qwen2.5-VL-7B-Instruct`
[7]`https://github.com/mjpost/sacrebleu`
[8]`https://github.com/QwenLM/Qwen2.5-VL`

---

**Instruction-Specific Rubrics Generation Prompt**

You are provided with a **video (or a detailed description)**, a corresponding **instruction**, and a **reference response**. The instruction defines a **video understanding task**, which may involve **open-ended question answering**, **captioning**, **instruction following**, **temporal reasoning**, or **multi-step inference grounded in the video**. Such tasks are open-ended and often require complex or nuanced reasoning over the visual and temporal content.

Your task is to **generate an instruction-specific evaluation rubric**.
The rubric will be used to rate the quality of any response to the instruction on a **1–5 scale**.
The **reference response is provided only to help you define what a perfect answer (Rating 5) looks like**, but during evaluation the rubric must stand alone — evaluators will not be given the reference response.

**Important Note:**
Read the provided **video description as if you are watching the video yourself**.
Pay close attention to all details in the video description and the reference response.
Use these to construct precise, example-specific scoring rubrics that can be applied without needing the reference later.

**Task:**
For the given video, instruction, and reference response:
- Generate a **single 1–5 scoring rubric** tailored to this instruction.
- Each score level (1 through 5) must include a clear, example-specific description of what a response at that level would look like.
- Use the **reference response** to anchor what counts as a "5 (Excellent)" answer.
- Ensure that the rubric is **self-contained** so that it can be applied without access to the reference response.

**Rating Guidelines:**
- **Rating 1 (Very Poor):** Completely wrong, irrelevant, or missing.
- **Rating 2 (Poor):** Vague, incomplete, or largely inaccurate with minimal grounding.
- **Rating 3 (Fair):** Partially correct, captures some aspects but misses key details.
- **Rating 4 (Good):** Mostly correct and grounded, covers most important aspects but not fully comprehensive.
- **Rating 5 (Excellent):** Fully correct, detailed, coherent, and well-grounded — aligned with the reference response.

**Input:**
**Video Content (as detailed description):**

`{video_description}`

**Instruction:**

`{instruction}`

**Gold Standard Reference Response:**

`{reference_response}`

**Output (Strict Format):**

```
**Rubric (Scale 1{5):**
- **Rating 1 (Very Poor):** [instruction-specific description]
- **Rating 2 (Poor):** [instruction-specific description]
- **Rating 3 (Fair):** [instruction-specific description]
- **Rating 4 (Good):** [instruction-specific description]
- **Rating 5 (Excellent):** [instruction-specific description]
```

Figure 10: LLM-as-judge prompt for generating instruction-specific rubrics for video understanding tasks.

**Human Evaluation of Generated Rubrics**  We conduct a human evaluation study on Amazon Mechanical Turk to compare the quality of rubrics generated for video–instruction–response triplets. The evaluation dataset consists of 300 randomly sampled rubrics for 6 models, including VideoJudge-3B (which we train). For each example, we generated rubrics using VideoJudge as one option and compared them against rubrics produced by five other models: GPT-4-mini, Qwen-3B, Qwen-7B, Qwen-32B, and Qwen-72B. Annotators were given the video along with its description, the instructions to the models, a reference response illustrating a good answer, and two candidate rubrics (A and B), each defined on a 1–5 scale. Their task was to compare the two rubrics and select the one they considered more effective for evaluating AI-generated responses to the given instruction. Each rubric pair was assessed independently by three annotators. The full annotation framework, including the instructions provided and a representative Human Intelligence Task (HIT) example, is shown in Figure 18.

**Decoding Temperature**  We provide other metrics for decoding at different temperature in Table 19.

Table 5: Rubrics generated by different models for the instruction *"What is the man wearing while climbing the rock?"*.

| Instruction | What is the man wearing while climbing the rock? |
|---|---|
| **Reference Response** | The man is wearing a grey west while climbing the rock. |

| Model | Rubric |
|---|---|
| Qwen2.5-VL-3B | **Rubric (Scale 1–5):** - **Rating 1 (Very Poor):** The response does not address the instruction at all or provides irrelevant information. - **Rating 2 (Poor):** The response partially addresses the instruction but contains significant inaccuracies or irrelevant details. - **Rating 3 (Fair):** The response addresses the instruction but lacks detail or is not entirely accurate. - **Rating 4 (Good):** The response addresses the instruction adequately and provides relevant information, but there are some inaccuracies or omissions. - **Rating 5 (Excellent):** The response addresses the instruction thoroughly and accurately, providing all necessary details and information. |
| Qwen2.5-VL-7B | **Rubric (Scale 1–5):** - **Rating 1 (Very Poor):** The response does not mention the man climbing the rock and instead describes a different scene or activity. - **Rating 2 (Poor):** The response mentions the man climbing the rock but includes irrelevant details or descriptions that do not match the video content. - **Rating 3 (Fair):** The response accurately describes the man climbing the rock but may include minor inaccuracies or omissions. - **Rating 4 (Good):** The response accurately describes the man climbing the rock and includes relevant details, but there might be some minor inaccuracies or omissions. - **Rating 5 (Excellent):** The response accurately describes the man climbing the rock, includes all relevant details, and does not contain any inaccuracies or omissions. |
| Qwen2.5-VL-32B | **Rubric (Scale 1–5):** - **Rating 1 (Very Poor):** The response is completely unrelated to the video content and does not mention any clothing or climbing gear. It may describe an entirely different scene or activity. - **Rating 2 (Poor):** The response mentions some clothing or gear, but it is incorrect or irrelevant to the video. For example, it might describe a bright red helmet or a beach setting, which are not present in the video. - **Rating 3 (Fair):** The response partially describes the clothing or gear worn by the man climbing the rock, but it is incomplete or contains minor inaccuracies. For example, it might mention a shirt and shorts but fail to note the climbing harness or shoes. - **Rating 4 (Good):** The response accurately describes most of the clothing and gear worn by the man climbing the rock, including a shirt, shorts, climbing harness, and shoes. However, it may miss minor details or fail to mention the climbing rope. - **Rating 5 (Excellent):** The response provides a complete and accurate description of all the clothing and gear worn by the man climbing the rock, including a shirt, shorts, climbing harness, shoes, and the climbing rope. It is precise and fully grounded in the video content. |
| Qwen2.5-VL-72B | **Rubric (Scale 1–5):** - **Rating 1 (Very Poor):** The response does not mention any clothing or equipment the man is wearing while climbing the rock. It fails to describe the climbing activity and instead provides irrelevant details about a beach scene. - **Rating 2 (Poor):** The response mentions some clothing or equipment but includes significant inaccuracies or irrelevant details. It may describe the wrong activity or setting, such as sitting on a rock or being on a beach. - **Rating 3 (Fair):** The response partially describes the man's attire while climbing but misses key details. It may mention some correct elements but lacks completeness or accuracy in describing the climbing gear and setting. - **Rating 4 (Good):** The response accurately describes the man's attire, including a tank top, shorts, and climbing shoes, while climbing the rock. It may also mention the climbing harness and rope but could be slightly lacking in detail or context. - **Rating 5 (Excellent):** The response comprehensively and accurately describes the man's attire, including a tank top, shorts, climbing shoes, a harness, and a rope, while climbing the rock. It provides a clear and detailed account of the climbing activity and setting, matching the video content perfectly. |
| GPT-4o-mini | **Rubric (Scale 1–5):** - **Rating 1 (Very Poor):** The response completely fails to address the instruction, providing irrelevant information about the climbing scene or omitting any mention of the man's clothing. - **Rating 2 (Poor):** The response offers vague or inaccurate information about the man's clothing, such as a general description (e.g., "he is wearing something") but fails to specify or properly identify the clothing item. - **Rating 3 (Fair):** The response provides a partially correct description of the man's clothing but may miss critical details, such as the specific color or type of clothing, leading to an incomplete answer (e.g., "the man is wearing a shirt"). - **Rating 4 (Good):** The response accurately identifies the man's clothing with most important details provided, such as the color and type of the clothing, but might lack some additional context or minor details (e.g., "the man is wearing a grey tank top"). - **Rating 5 (Excellent):** The response is fully correct, detailed, and coherent, clearly stating that the man is wearing a grey tank top while climbing the rock, and aligns perfectly with the instruction. |
| VideoJudge-3B | **Rubric (Scale 1–5):** - **Rating 1 (Very Poor):** The response completely misses the question about what the man is wearing while climbing, providing irrelevant information or no answer at all. - **Rating 2 (Poor):** The response gives a vague or incomplete answer regarding the man's clothing, such as mentioning he is "dressed" or "wearing something," but lacks specific details or accuracy. - **Rating 3 (Fair):** The response provides some correct information about the man's clothing but misses key details, such as stating he is "wearing shorts" without specifying the color or type of clothing. - **Rating 4 (Good):** The response accurately describes the man's clothing while climbing, mentioning specific items like "a tank top and shorts," but may lack additional context or detail about his overall appearance. - **Rating 5 (Excellent):** The response fully and accurately describes the man's clothing while climbing, clearly stating "the man is wearing a tank top and shorts," and may also include additional relevant details about his appearance or actions. |

**Number of Frames** We provide other metrics for max frames ablation in Figure 20.

---

**Evaluating Model-Generated Responses Prompt**

You are provided with a **video**, a corresponding **instruction**, and a **response** generated by a model. The instruction defines a **video understanding task**, which may take any form — including but not limited to **open-ended question answering**, **captioning**, **instruction following**, **temporal reasoning**, or **multi-step inference grounded in the video**. These tasks are open-ended and often require complex or nuanced reasoning over the visual and temporal content.

Your task is to **evaluate the quality of the response**, considering how well it satisfies the task defined by the instruction, based on the content of the video. This is a holistic judgment and should be based on the overall **correctness**, **relevance**, **completeness**, and **grounding** of the response.

**Task:**
For each response:
- Assess how well it addresses the task in the instruction in the context of the video.
- Consider whether the response is accurate, relevant, complete, and grounded in the video.
- Provide a brief rationale explaining the overall quality and alignment of the response with the instruction inside `<thinking></thinking>`.
- Output a score from 1 (worst) to 5 (best) indicating the overall quality inside `<score> </score>`.

**Rating Guidelines:**
- **Rating 5**: Fully accurate, complete, and well-grounded. The response precisely follows the instruction with no notable errors or omissions.
- **Rating 4**: Mostly accurate with minor issues. The response preserves the key meaning and relevance, with only small lapses in detail or precision.
- **Rating 3**: Partially correct. The response conveys the general idea but includes noticeable errors, omissions, or misinterpretations.
- **Rating 2**: Poor alignment. The response has serious flaws and only loosely relates to the instruction or video.
- **Rating 1**: Unrelated or incorrect. The response fails to reflect the video or follow the instruction meaningfully.

**Input:**
**Instruction:**

`{instruction}`

**Response:**

`{response}`

**Output (Strict Format):**

```
<thinking>{{your reasoning and explanation for the rating}}</thinking>
<score>{{integer score from 1 to 5}}</score>
```

Figure 11: LLM-as-judge prompt for evaluating model-generated responses to video understanding tasks.

---

**Algorithm 1:** Bootstrapping Training Data with Self-Refinement

---

**Input:** Video $v$, video description $\tilde{v}$, instruction $x$, gold response $y^*$, generator $G$, evaluator $E$, threshold $\alpha$, max iterations $T$

**Output:** Bootstrapped dataset $\mathcal{D}$

Initialize $\mathcal{D} \leftarrow \{(v, x, y^*, N)\}$ // gold response with max rating

**for** $r \in \{1, \ldots, N-1\}$ **do**

    $y_0^{(r)} \leftarrow G(p_{\text{gen}}\|\tilde{v}\|x\|y^*, r)$ // initial generation

    **for** $t \in \{0, \ldots, T-1\}$ **do**

        // feedback and refinement

        $\hat{r}, f_t^{(r)} \leftarrow E(p_{\text{eval}}\|\tilde{v}\|x\|y^*\|y_t^{(r)})$

        **if** $|r - \hat{r}| \leq \alpha$ **then**

            $\mathcal{D} \leftarrow \mathcal{D} \cup \{(v, x, y_t^{(r)}, r)\}$

            **break**

        **else**

            $y_{t+1}^{(r)} \leftarrow G(p_{\text{ref}}\|\tilde{v}\|x\|y^*\|y_t^{(r)}\|f_t^{(r)}, r)$

**return** $\mathcal{D}$

---

---

### Rubric Generation and Response Evaluation Prompt

You are provided with a **video**, a corresponding **instruction**, and a **response** generated by a model. The instruction defines a **video understanding task**, which may take any form — including but not limited to **open-ended question answering**, **captioning**, **instruction following**, **temporal reasoning**, or **multi-step inference grounded in the video**. These tasks are open-ended and often require complex or nuanced reasoning over the visual and temporal content.

Your task has two parts:

**Part 1 — Rubric Generation:**
- Generate an **instruction-specific evaluation rubric** on a **1–5 scale** for this example.
- The rubric must define what a Rating 1 through Rating 5 response would look like **in the context of this exact video and instruction**.
- The rubric should be **tailored to this instruction** and **grounded in the video content**, not generic.
- Each rating should describe concrete aspects of the video/instruction that would or would not appear in a response at that level.
- The rubric should be **self-contained** so it can later be applied to any response to the same instruction, without needing additional references.
- Output your rubric within `<rubric></rubric>`.

**Part 2 — Response Evaluation:**
- Using the rubric you just generated, evaluate the provided response.
- Assess how well it addresses the task in the instruction in the context of the video.
- Consider whether the response is accurate, relevant, complete, and grounded in the video.
- Provide a brief rationale explaining the overall quality and alignment of the response with the instruction inside `<thinking></thinking>`.
- Output a score from 1 (worst) to 5 (best) indicating the overall quality inside `<score> </score>`.

**Input:**
**Instruction:**

`{instruction}`

**Response:**

`{response}`

**Output (Strict Format):**

```
<rubric>
**Rubric (Scale 1{5):**
- **Rating 1 (Very Poor):** [instruction-specific description referencing video details]
- **Rating 2 (Poor):** [instruction-specific description referencing video details]
- **Rating 3 (Fair):** [instruction-specific description referencing video details]
- **Rating 4 (Good):** [instruction-specific description referencing video details]
- **Rating 5 (Excellent):** [instruction-specific description referencing video details]
</rubric>
<thinking>{{your reasoning and explanation for the rating}}</thinking>
<score>{{integer score from 1 to 5}}</score>
```

Figure 12: LLM-as-judge prompt for generating rubrics and evaluating responses to video understanding tasks.

---

**Rubric Comparison Prompt**

You are provided with a **video**, a corresponding **instruction**, and a **reference response**. The instruction defines a **video understanding task**, which may take any form — including but not limited to **open-ended question answering**, **captioning**, **instruction following**, **temporal reasoning**, or **multi-step inference grounded in the video**. These tasks are open-ended and often require complex or nuanced reasoning over the visual and temporal content.

You are also provided with **two rubrics (Rubric A and Rubric B)**, each generated by a model. These rubrics are intended to evaluate responses to the instruction on a **1–5 scale**.

Your task is to **decide which rubric is better** for evaluating responses to the instruction. This is a holistic judgment and it should be based on the rubric's **specificity**, **clarity**, **coverage**, and **usefulness for evaluation**.

**Task:**
For each pair of rubrics:
- Assess which rubric better reflects the instruction and video content (instruction-specificity).
- Consider whether the rubric is self-contained and usable without the reference response.
- Evaluate the clarity, distinctness, and logical progression of the rating levels (1–5).
- Judge which rubric provides better coverage of key aspects required to evaluate responses.
- Provide a brief rationale explaining your choice inside `<thinking> </thinking>`.
- Output the preferred rubric as A or B inside `<answer> </answer>`.

**Guidelines for Comparison:**
- Prefer the rubric that is **more specific** to the given instruction and video.
- Prefer the rubric that is **clear, well-structured, and easy to apply**.
- Prefer the rubric that **captures all important aspects** of what makes a good or bad response.
- If both rubrics are strong, choose the one that is **slightly more precise or comprehensive**.
- Do not output a tie — always select either A or B.

**Input:**
**Instruction:**
`{instruction}`

**Reference Response:**
`{ref_response}`

**Rubric A:**
`{rubric_a}`

**Rubric B:**
`{rubric_b}`

**Output (Strict Format):**

```
<thinking>{{your reasoning and explanation for why one rubric is better}}</thinking>
<answer>{{A or B}}</answer>
```

Figure 13: LLM-as-judge prompt for pairwise comparison of two instruction-specific rubrics.

---

**Pairwise Response Comparison Prompt**

You are provided with a **video**, a corresponding **instruction**, and **two candidate responses** generated by two models. The instruction defines a **video understanding task**, which may involve **open-ended question answering**, **captioning**, **instruction following**, **temporal reasoning**, or **multi-step inference grounded in the video**. Such tasks are open-ended and often require complex or nuanced reasoning over the visual and temporal content.

Your task is to **compare the two responses** and decide which one better addresses the instruction, based on the content of the video. Make a holistic judgment that considers **correctness**, **relevance**, **completeness**, and **grounding**.

**Task:**
For the given pair of responses:
- Judge which response better addresses the instruction in the context of the video.
- Consider whether each response is accurate, relevant, complete, and grounded in the video.
- Output only A or B, wrapped strictly inside `<answer></answer>` tags.

**Evaluation Guidelines:**
- **Accuracy**: Prefer responses that are factually correct and consistent with the video.
- **Relevance**: Prefer responses that directly answer the instruction without digression.
- **Completeness**: Prefer responses that capture all key aspects needed for a full answer.
- **Grounding**: Prefer responses clearly supported by the video, avoiding hallucinations.
- If one response contains hallucinations, irrelevant content, or omissions, prefer the other.
- If both responses are strong, choose the one that is more precise and detailed.
- If both responses are weak, choose the one that is less flawed.

**Input:**
**Instruction:**

`{instruction}`

**Response A:**

`{response_a}`

**Response B:**

`{response_b}`

**Output (Strict Format):**

```
<answer>{{A_or_B}}</answer>
```

Figure 14: LLM-as-judge prompt for pairwise comparison of model-generated responses to video understanding tasks.

---

**Pairwise Response Evaluation with Reasoning Prompt**

You are provided with a **video**, a corresponding **instruction**, and **two candidate responses** generated by two models. The instruction defines a **video understanding task**, which may involve **open-ended question answering**, **captioning**, **instruction following**, **temporal reasoning**, or **multi-step inference grounded in the video**. Such tasks are open-ended and often require complex or nuanced reasoning over the visual and temporal content.

Your task is to **compare the two responses** and decide which one better addresses the instruction, based on the content of the video. Make a holistic judgment that considers **correctness**, **relevance**, **completeness**, and **grounding**.

**Task:**
For the given pair of responses:
- First, evaluate **Response A** (strengths and weaknesses).
- Then, evaluate **Response B** (strengths and weaknesses).
- Finally, compare them and decide which one better satisfies the instruction in the context of the video.
- Ground your reasoning in the video content — treat it as if you are directly watching the video.
- Write the reasoning inside `<thinking></thinking>`.
- After the reasoning, output only A or B, wrapped strictly inside `<answer></answer>` tags.

**Evaluation Guidelines:**
- **Accuracy**: Prefer responses that are factually correct and consistent with the video.
- **Relevance**: Prefer responses that directly answer the instruction without digression.
- **Completeness**: Prefer responses that capture all key aspects needed for a full answer.
- **Grounding**: Prefer responses clearly supported by the video, avoiding hallucinations.
- If one response contains hallucinations, irrelevant content, or omissions, prefer the other.
- If both responses are strong, choose the one that is more precise and detailed.
- If both responses are weak, choose the one that is less flawed.

**Input:**
**Instruction:**

`{instruction}`

**Response A:**

`{response_a}`

**Response B:**

`{response_b}`

**Output (Strict Format):**

```
<thinking>
{{Evaluate Response A -> Evaluate Response B -> Compare them concisely,
explaining why one is better, grounded in the video.}}
</thinking>
<answer>{{A_or_B}}</answer>
```

Figure 15: LLM-as-judge prompt for pairwise evaluation of responses with explicit stepwise reasoning.

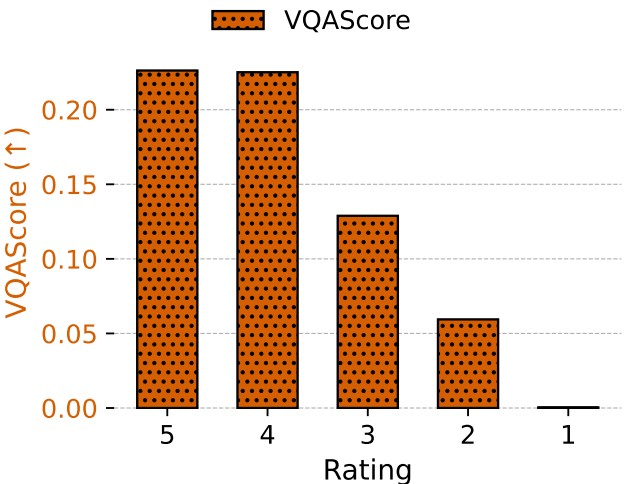

Figure 16: Absolute VQAScores across specified ratings (1–5). Higher-rated responses achieve higher scores, showing alignment between VQA score and specified ratings.

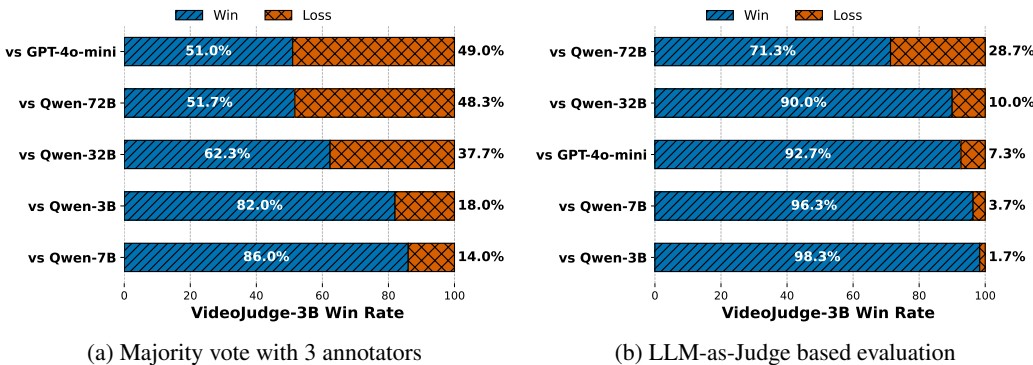

(a) Majority vote with 3 annotators

(b) LLM-as-Judge based evaluation

Figure 17: Win rate of **VideoJudge-3B** compared to other models under two evaluation settings. 17a: majority voting of human annotator, where the most common annotators' choice determines the label. 17b: LLM-as-Judge preference with deterministic decoding ($T = 0$). Across both settings, **VideoJudge-3B** consistently produces high-quality rubrics and achieves performance competitive with, or surpassing, models up to $25\times$ larger, including proprietary systems such as GPT-4o-mini.

Table 6: Representative examples of video frames paired with instructions and bootstrapped responses at different rating levels (R1–R5) generated by our pipeline.

| Video Frames | Instruction and Responses |
|---|---|
| 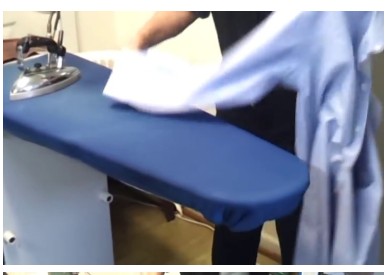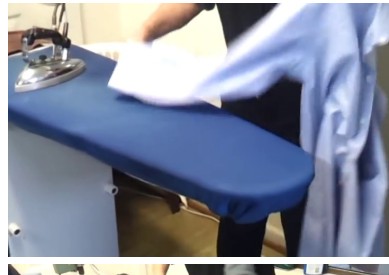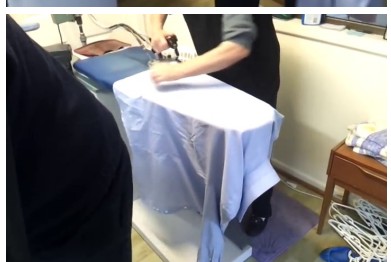 | **Instruction:** What is the man wearing while ironing the dress shirt?

**R5:** The video shows a white building from the outside, the inside of a ballet studio where children practice, a girl speaking from inside a well-lit studio, children walking through the school, and a boy and a girl dancing in a studio as other children watch.

**R4:** The video features an exterior view of a grand building, followed by scenes inside a ballet studio where young dancers practice. There are moments showing a girl speaking and other children walking through the school, along with a boy and a girl performing ballet moves while others observe.

**R3:** The video features a beautiful building at the start, then transitions into a ballet class where children are practicing their routines. A girl talks at one point, and there are some glimpses of kids walking around. The main focus is on the ballet practice, with some children watching and learning.

**R2:** The video shows a building and then cuts to some kids who are doing ballet in a studio. There is a girl who talks briefly, and some kids are just wandering around. The dancing is mixed with scenes of children doing other activities, which makes it hard to tell what's happening.

**R1:** The video features a beautiful building at the start, then transitions into a ballet class where children are practicing their routines. A girl talks at one point, and there are some glimpses of kids walking around. The main focus is on the ballet practice, with some children watching and learning. |
| 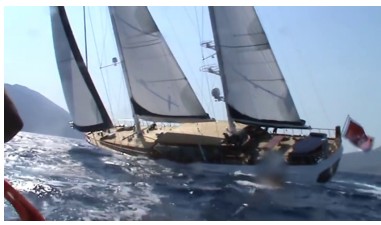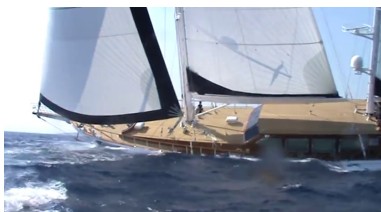 | **Instruction:** Can you describe in detail what happens in the video?

**R5:** The video depicts a ship in the sea during the daytime, with a hill visible in the distance at the shore. The ship is dangerously tilted towards the left and appears to be on the verge of capsizing, while all its sails are fully deployed. Another group of people can be seen approaching the ship in a motorboat. The video ends with the camera panning around the motorboat and the surrounding area. There are no visible signs of distress or emergency response. Overall, the video captures a dramatic and potentially hazardous situation on the high seas.

**R4:** The video shows a sailing yacht navigating through rough waters during the day, with a distant coastline visible. The yacht is leaning significantly to one side, giving the impression that it might capsize, and all its sails are up. A motorboat approaches the yacht, but there are no signs of distress. The scene captures an adventurous sailing experience in open waters.

**R3:** The video features a yacht sailing in the ocean on a sunny day, with some hills visible far in the background. The yacht is leaning to one side, which could look dangerous. All the sails are fully deployed, and it appears to be a thrilling experience. There seems to be no immediate danger, and the overall atmosphere feels calm despite the tilt of the yacht, making it look like a leisurely outing rather than an emergency.

**R2:** The video presents a boat moving across the water, and it's bright outside. There are some distant landforms, possibly hills. The boat seems to be leaning, which might suggest it's having trouble. A small boat is around, but it's unclear what the situation is. The video feels like it's capturing a sailing adventure.

**R1:** The video shows a large ship floating in calm waters, with people enjoying a picnic on the deck. In the background, there's a beautiful sunset, and the ship appears to be stationary. The focus is on people laughing and eating, with no signs of sailing or any movement. |

Table 7: Pairwise human evaluation results across two annotators. The table reports overall inter-annotator agreement and Cohen's Kappa as measures of reliability. Preference distributions show the proportion of times each annotator selected response b versus response a, indicating a slight bias toward b. Correctness is computed as the fraction of instances where the annotator's preferred response matches the higher-rated (gold) response, with both annotators showing high accuracy over 250 samples each. The last two rows capture error analysis: in 11 cases (4.4%), both annotators agreed on an incorrect answer, while in 13 cases (5.2%), the annotators disagreed on a correct answer, highlighting residual uncertainty.

| Metric | Value |
|---|---|
| Agreement | 94.80 |
| Cohen's Kappa | 89.54 |
| Annotator1 Preference (b / a) | 53.2 / 46.8 |
| Annotator2 Preference (b / a) | 54.4 / 45.6 |
| Annotator1 Correctness | 92.40 (250 samples) |
| Annotator2 Correctness | 93.60 (250 samples) |
| Both agreed on wrong answers | 11 / 250 (4.4) |
| Both disagreed on correct answers | 13 / 250 (5.2) |

Table 8: Comparison of annotator decisions across instruction–response pairs.

| Instruction | Response A | Response B | A1 | A2 |
|---|---|---|---|---|
| What is happening in the video? | The video shows a couple dancing outside, with some people watching them. Children are playing in the background, and there might be someone cooking food. Another couple appears later, but the details are a bit unclear. | In the video, a couple is moving around in what looks like a backyard gathering. There are kids playing, and it seems like a party. A few adults are standing around, but it's hard to see what's really happening. | A | A |
| Can you describe the video in detail? | In the video, a man in a camouflage shirt is seen doing nail care for a woman. He appears to be using some kind of product, and there are people sitting nearby. The lighting seems good, and the focus is on the nails, but there are moments where it shifts away from the main action. | The video features a casual setting where a person is doing something with nails. There is a man in a camouflage shirt, and it looks like he is applying some kind of treatment. Other people are around, but the details about the process are quite vague and unclear. | B | A |
| Can you describe the competition the man is participating in? | The video shows a man participating in a sports competition where he throws a round object in a circular area. He attempts to throw it a long distance, and there are some spectators watching, but the focus on the throwing technique is not very clear, and the details about the event are vague. | In the video, a man is seen throwing a discus in a competition. He spins to gain speed and then throws it. The footage captures him from different perspectives, and there are some people in the crowd, but not many details about them. | A | B |
| What is the woman wearing and what is behind her? | The woman has on a green shirt, and there seems to be a clothing item behind her, but it's not specific. The room is bright, and there might be some products nearby, but the details are unclear. | The woman is wearing a green top, and behind her is a denim jacket. She is in a well-lit room with a window, and there are some products on a table, but it's not clear what specific items they are. | B | B |

Table 9: Training and evaluation hyperparameters.

| Hyperparameter | Value |
|---|---|
| *Training* | |
| Learning rate | 2e-7 |
| Batch size (per device) | 16 |
| Gradient accumulation steps | 1 |
| Num. train epochs | 2 |
| Warmup ratio | 0.03 |
| Weight decay | 0.0 |
| Max grad norm | 1.0 |
| LR scheduler | Cosine |
| Precision | bfloat16 |
| Max sequence length | 128,000 |
| Video max frames | 60 |
| Video max frame pixels | 25,088 |
| Video min frame pixels | 3,136 |
| Gradient checkpointing | True |
| tune_mm_vision | False |
| tune_mm_mlp | True |
| tune_mm_llm | True |
| *Evaluation* | |
| max_new_tokens | 1024 |
| fps | 1 |
| max_frames | 180 |
| max_pixels | $20480 \times 28 \times 28$ |
| min_pixels | $16 \times 28 \times 28$ |

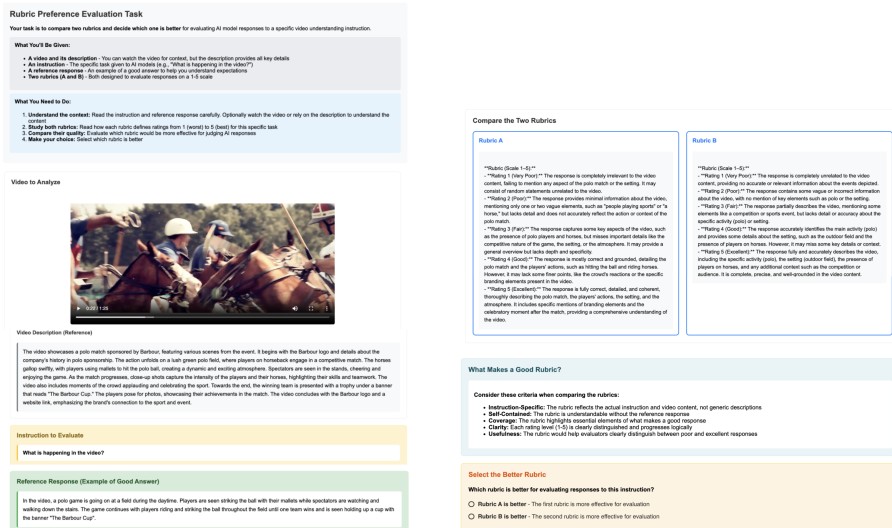

Figure 18: Example of the Human Evaluation MTurk Interface. Annotators were provided with a video, its description, the instruction, a reference response, and two candidate rubrics. They compared the rubrics and selected the one they considered more effective for evaluating AI-generated responses.

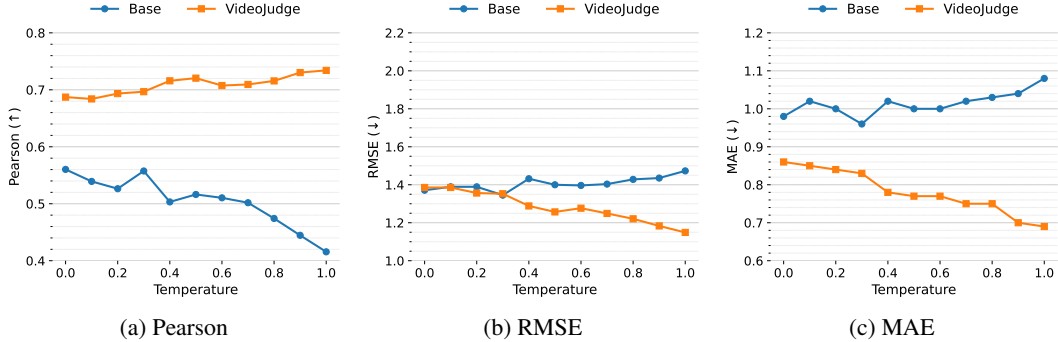

(a) Pearson       (b) RMSE       (c) MAE

Figure 19: Effect of decoding temperature on pointwise judgment reliability for the base Qwen2.5-VL-3B and the VideoJudge-3B model, evaluated using Pearson correlation, RMSE, and MAE. As temperature increases, the base model shows steady degradation in correlation and higher error rates, indicating sensitivity to stochastic decoding. In contrast, the VideoJudge model remains stable and even improves at moderate to high temperatures, achieving higher correlation and lower error. This suggests that rubric-guided training yields substantially more temperature-robust and consistent judgments under non-deterministic decoding.

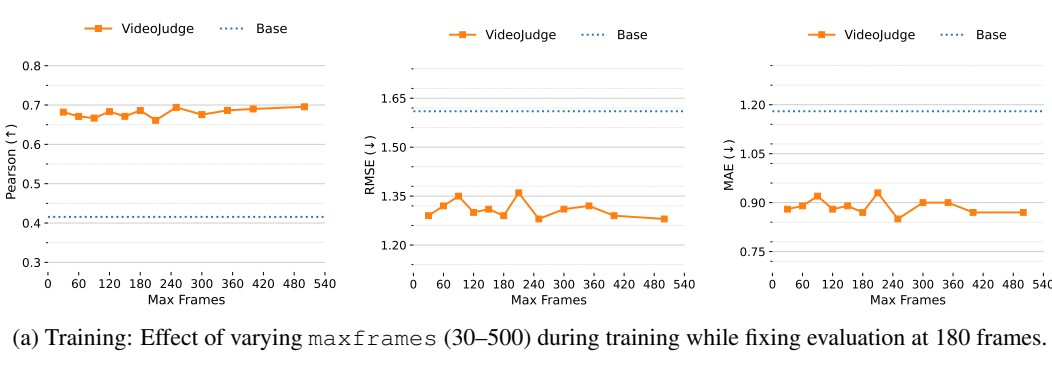

(a) Training: Effect of varying `maxframes` (30–500) during training while fixing evaluation at 180 frames.

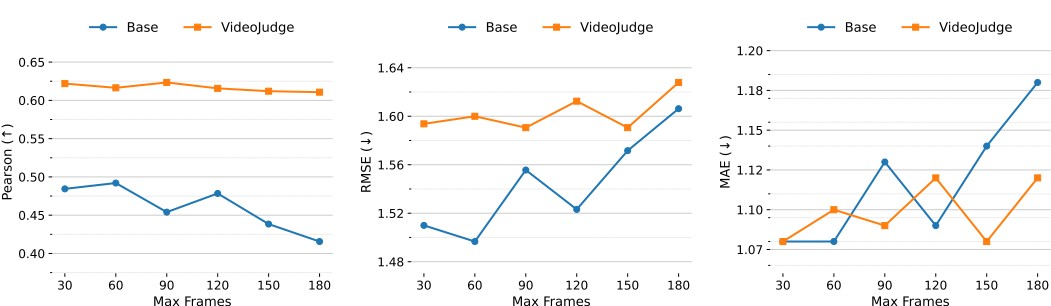

(b) Evaluation: Effect of varying `maxframes` (30–180) at inference while fixing training at 60 frames.

Figure 20: Effect of temporal context (`maxframes`) on VideoJudge performance across Pearson, RMSE, and MAE. Increasing temporal coverage during training improves correlation and reduces error up to ∼240 frames, after which gains plateau. At evaluation time, performance improves up to ∼120 frames and then saturates, indicating diminishing returns beyond moderate temporal context.

