# OpenReview forum: "VideoJudge: Bootstrapping Enables Scalable Supervision of MLLM-as-a-Judge for Video Understanding"
_ICLR.cc/2026/Conference — ICLR 2026 Poster_

### Official Review · Reviewer_zfmP · 2025-10-24

**Soundness:** 3
**Presentation:** 2
**Contribution:** 2
**Rating:** 4
**Confidence:** 2

**Summary:**

This paper proposes VideoJudge, a bootstrapped framework for training multimodal large language models to serve as automatic evaluators for video understanding tasks. By generating synthetic data through an iterative generator–evaluator process, the method enables scalable supervision without human labels. The resulting small models outperform larger MLLM judges in correlation with human judgments across multiple benchmarks.

**Strengths:**

1.	The generator-evaluator cycle to produce synthetic training data for the judge is meaningful and well-motivated. It reduces reliance on human annotations and enables larger scale.

2.	The authors show that a relatively small model (3B) can outperform much larger baselines when trained appropriately, which is a strong practical result.

**Weaknesses:**

1.	The bootstrapping relies on LLM-based generator and evaluator and that LLM may carry bias and errors. If the evaluator is weak, the whole pipeline could propagate flawed judgments. The paper could more deeply analyze potential bias or drift in this synthetic supervised signal.

2.	While the benchmarks used show strong correlation results, it is not entirely clear how robust the judge will be to entirely new video domains, tasks, or instruction types. The paper would benefit from a “domain shift” evaluation (unseen video types).

3.	While the correlation metrics are good, more detailed breakdowns of failure cases where the judge disagrees with humans would help understand the limitations. Are there types of errors the judge misses (e.g., subtle temporal reasoning, common sense)?

**Questions:**

See weaknesses.

---

> ### Author Response · Authors · 2025-11-27
> **Rebuttal -- Reviewer zfmP (1/2)**
>
> We thank Reviewer zfmP for the thoughtful feedback. The concerns about bias propagation, domain shift, and failure-case analysis are central to evaluating any MLLM-based judge, and we appreciate the opportunity to clarify these points. As detailed below, VideoJudge surpasses its evaluator on multiple independent human-annotated benchmarks, generalizes well across diverse video tasks and domains, and shows clear patterns in both its strengths and failure modes. These results directly address the reviewer’s concerns and provide a clearer picture of the model’s reliability and limitations.
>
>
> ### **W1. Addressing Bias and Error Propagation in Bootstrapping**
> >The bootstrapping relies on LLM-based generator and evaluator and that LLM may carry bias and errors. If the evaluator is weak, the whole pipeline could propagate flawed judgments. The paper could more deeply analyze potential bias or drift in this synthetic supervised signal.
>
>
>
> We appreciate the reviewer’s concern about potential bias or error amplification in the bootstrapped generator–evaluator pipeline. Empirically, however, VideoJudge-7B does not inherit the evaluator’s weaknesses. If it were merely imitating the zero-shot Qwen2.5-VL-7B evaluator, its performance would be bounded by that model. Instead, it consistently surpasses the evaluator across benchmarks—for example, VATEX RMSE improves from 2.36 → 1.46, and LongVideoBench Δ(C–D) increases from 0.35 → 1.16. VideoJudge-7B also matches or exceeds much larger models (e.g., Qwen2.5-VL-72B), indicating that it learns generalizable evaluation behavior rather than reproducing evaluator-specific biases. Human validation further supports this, with 92–94% correctness and high inter-annotator agreement.
>
> Our bootstrapping framework additionally includes bias-mitigation steps: we reject and regenerate samples where evaluator ratings diverge from intended targets, preventing noisy supervision; we draw seed data from multiple heterogeneous video–instruction sources to avoid stylistic overfitting; and the model’s strong performance on human-annotated datasets such as VATEX-Eval demonstrates that it does not propagate evaluator errors. While synthetic supervision is imperfect, these mechanisms—together with consistent gains on independent human benchmarks—show that the pipeline effectively limits bias drift and yields a reliable, human-aligned video judge.
>
> ### **W2. Robustness to Domain Shift and Unseen Video Types**
> >While the benchmarks used show strong correlation results, it is not entirely clear how robust the judge will be to entirely new video domains, tasks, or instruction types. The paper would benefit from a “domain shift” evaluation (unseen video types).
>
>
>
> We appreciate the reviewer’s suggestion and agree that robustness to domain shift is crucial for a reliable video evaluator. While we did not include an explicit “domain shift” section, our current evaluation already spans substantial variation in video types, tasks, and annotation sources. The training data (VideoInstruct-100K, VCG-Plus-112K, VideoChat2-IT) differs significantly from our evaluation benchmarks, which include LLaVA-Video instructions (open-ended QA), VideoChatGPT-style descriptions, multilingual human-annotated VATEX scores, long-duration temporal reasoning videos in LongVideoBench (up to 297 s), and real-world human preference data from VideoAutoArena. VideoJudge-7B achieves strong and consistent performance across all these settings—e.g., 0.78 correlation on VideoJudgeLLaVA, 85.49% accuracy on VideoAutoArena, 1.46 RMSE on VATEX, and Δ(C–D) = 1.16 on LongVideoBench—despite their distinct distributions and task formats, indicating meaningful generalization beyond the training domain.
>
> We also highlight mechanisms that support robustness to unseen domains: VideoJudge surpasses text-only baselines on VATEX and LongVideoBench, showing that it leverages video signals rather than relying on textual shortcuts; and the rubric-generation module (VideoJudgeR-3B) produces instance-specific evaluation criteria, enabling adaptive reasoning across task types. We acknowledge that extreme domain shifts (e.g., medical or scientific videos, specialized activities, non-English instructions) are not covered in our current benchmarks. In the revised manuscript, we will include additional domain-shift analyses—such as zero-shot evaluation on ActivityNet-like datasets and subgroup analyses by video duration and task type—and expand our discussion of limitations. Nonetheless, the strong cross-benchmark performance already demonstrates substantial robustness to distributional and task-level shifts.

---

> ### Author Response · Authors · 2025-11-27
> **Rebuttal -- Reviewer zfmP (2/2)**
>
> ### **W3. Analysis of Failure Cases and Error Types**
>
> >While the correlation metrics are good, more detailed breakdowns of failure cases where the judge disagrees with humans would help understand the limitations. Are there types of errors the judge misses (e.g., subtle temporal reasoning, common sense)?
>
> Thank you for the valuable feedback. We have conducted a comprehensive error analysis examining the 5000 evaluation by our judge models. Here are our key findings on the types of errors the judge misses:
>
> **Overestimation Bias** Our analysis reveals a  bias toward overestimation: the judge overestimates by ≥2 points in 14.8% of cases but underestimates by the same margin in only 1.5% of cases (9.8x ratio). This indicates the model is consistently too lenient rather than too harsh.
>
> Primary Error Types
> 1. Vague and Hedging Language (47.9% of overestimations)
> The judge fails to penalize responses that use hedging language like "seems to be," "appears to," "looks like," or "some kind of." Rather than recognizing these as indicators of uncertainty or lack of knowledge, the judge treats vague descriptions as acceptable responses.
> Example: A response stating "The gymnast is in casual clothes, like a t-shirt and jeans" (factually incorrect) received a score of 5/5, with the judge praising it for "accurately identifying" the attire.
> 2. Lack of Specificity (29.5% of overestimations)
> The judge often misses when responses provide general descriptions instead of specific details requested in the instruction. It conflates topical relevance with accuracy, treating "the response is about the right topic" as equivalent to "the response correctly answers the question."
> 3. Temporal Reasoning Errors (10.6% of overestimations)
> The judge struggles with questions requiring understanding of sequence, chronological order, or temporal relationships. It often accepts generic descriptions when specific temporal ordering was requested, suggesting weak understanding of time-dependent video content.
> 4. Subtle Factual Errors (4.5% of overestimations)
> While the judge catches blatant factual errors (60% of underestimations involve clear inaccuracies), it misses subtle misalignments between the response and video content, particularly when the response is "close enough" or partially relevant.
> 5. Common Sense Failures
> The judge occasionally accepts responses that describe implausible or inconsistent scenarios (e.g., describing kitchen/cooking activity when video shows painting), suggesting limitations in common-sense verification against video content.
> Reasoning Pattern Analysis
> Examining the judge's evaluation text reveals a critical asymmetry:
>
> In overestimations: Positive-to-negative phrase ratio = 11.2:1 (2.13 vs 0.19 phrases)
> In accurate cases: Positive-to-negative phrase ratio = 1.1:1 (1.26 vs 1.13 phrases)
>
> The judge exhibits "accentuate the positive" bias, focusing heavily on what the response gets right while minimizing or overlooking errors. This explains why responses with partial correctness receive disproportionately high scores.
> Most Problematic Rating Scenarios
>
> Mid-quality responses (True Rating = 3): Only 36.9% exact agreement; 46.6% inflated to perfect scores of 5
> Near-perfect responses (True Rating = 4): Only 13.7% exact agreement; 81.3% upgraded to 5
> Poor responses (True Rating = 1): 12.6% scored as perfect 5/5 despite being lowest quality
>
> These patterns suggest the judge has difficulty distinguishing between "good" (4) and "perfect" (5), and struggles most with middle-range quality assessments where nuanced evaluation is required. The VideoJudge model exhibits the following limitations:
>
> 1. Does not adequately penalize vagueness in video descriptions
> 2. Weak temporal and sequential reasoning for time-dependent questions
> 3. Insufficient common-sense grounding to catch implausible descriptions
> 4. Asymmetric error detection: better at identifying blatant errors than subtle ones
> 5. Positive bias in reasoning: weighs correctness heavily, minimizes mistakes
> 6. Poor calibration between rating levels 3-5
>
> These findings suggest the need for training data emphasizing hard negatives, explicit vague-language detection, and improved temporal reasoning capabilities. We will include this along with error analysis for all models in the camera-ready version of our work.

---

### Official Review · Reviewer_Lbto · 2025-10-25

**Soundness:** 3
**Presentation:** 3
**Contribution:** 2
**Rating:** 4
**Confidence:** 4

**Summary:**

This paper addresses the challenge of scalable and reliable evaluation for video understanding tasks. The authors introduce VideoJudge, a 3B and 7B MLLM-based evaluator model trained via a bootstrapped generator-evaluator pipeline, where candidate responses at different quality levels are iteratively refined and filtered before being used as supervision. The system supports both pointwise rating and pairwise comparison, and the authors also construct meta-evaluation benchmarks to verify alignment with human preferences. Experiments show that VideoJudge models achieve comparable or even superior judgment quality to significantly larger MLLMs.

**Strengths:**

1. Overall, the paper is generally well-organized and clearly motivates the need for scalable video evaluation beyond standard reference-based metrics.
2. The proposed bootstrapping framework is reasonable and avoids costly manual annotation, which is an important practical benefit.
3. Experiments include multiple settings (pointwise, pairwise) and span several baseline models, showing consistent improvements.

**Weaknesses:**

1. While the paper claims this is "the first bootstrapped framework for training scalable MLLM-based evaluators across diverse video understanding tasks", it would benefit from clarifying how it differs from existing MLLM-as-a-Judge work that also uses distillation from other evaluators (e.g., VideoScore and Prometheus-Vision). The novelty statement could be toned down to avoid overclaim.
2. The performance gains over baseline on multiple datasets seem to be limited. According to Table 1, VideoJudge-7B performs worse than LLaVA-NeXT-7B (which shall be considered as a weak baseline given the weaker general video understanding capabilities than Qwen-2.5-VL-7B) under several metrics, yielding concerns regarding the actual effectiveness.
3. Besides, Table 1 should include more MLLM-as-a-Judge works to clearly identify the potential effectiveness and significance of the proposed scheme.

**Questions:**

Please refer to the weakness section for my questions. My major concerns are about the performance and the very limited baselines. Strong justifications shall be provided to demonstrate the significance of this submission.

Besides, the authors claim that "we have anonymously released the model checkpoints and datasets used for training and evaluation on HuggingFace", but the links are not found.

---

> ### Author Response · Authors · 2025-11-27
> **Rebuttal -- Reviewer Lbto (1/2)**
>
> We thank Reviewer Lbto for the thoughtful feedback. The points raised—regarding our distinction from prior MLLM-as-a-Judge work, interpretation of performance gains, and breadth of baselines—address core questions about the novelty and effectiveness of VideoJudge. We appreciate the opportunity to clarify these aspects. Below, we outline how VideoJudge differs in scope and methodology from prior frameworks, provide clearer evidence of its improvements over comparable baselines, and address the availability of model checkpoints and datasets. We believe these clarifications resolve the reviewer’s concerns and strengthen the contribution of our work.
>
> ### **W1. Distinguishing VideoJudge from Prior MLLM-as-a-Judge Work**
> >While the paper claims this is "the first bootstrapped framework for training scalable MLLM-based evaluators across diverse video understanding tasks", it would benefit from clarifying how it differs from existing MLLM-as-a-Judge work that also uses distillation from other evaluators (e.g., VideoScore and Prometheus-Vision). The novelty statement could be toned down to avoid overclaim.
>
> We appreciate the reviewer's comment to clarity regarding how VideoJudge differs from existing MLLM-as-a-Judge approaches. We acknowledge that our novelty statement could be more precise, and we clarify the key distinctions below.
>
> **Domain and Task Scope.** VideoScore (He et al., 2024) focuses exclusively on video generation evaluation, rating synthesized videos across dimensions like visual quality, temporal consistency, and text-to-video alignment. Prometheus-Vision (Lee et al., 2024b) extends LLM-as-a-Judge to image understanding, evaluating image captioning and VQA outputs. In contrast, VideoJudge targets video understanding tasks—including long-form video QA, temporal reasoning, event localization, and instruction following—which require models to assess whether a textual response accurately reflects complex spatiotemporal video content. This is a fundamentally different evaluation challenge: instead of rating generated videos or static images, VideoJudge must judge whether language outputs correctly interpret dynamic, temporally extended visual narratives.
>
> **Bootstrapping**. While VideoScore trains on human annotations of generated videos and Prometheus-Vision distills from GPT-4V's image-conditioned judgments, VideoJudge introduces a generator–evaluator bootstrapping loop that iteratively synthesizes and refines training data without requiring large-scale human annotation or direct distillation from proprietary models. This self-refinement loop yields over 100K training examples and enables VideoJudge-7B to exceed the judgment accuracy of models 10× larger (e.g., Qwen2.5-VL-72B) on video understanding benchmarks. Neither VideoScore nor Prometheus-Vision employ this iterative bootstrapping strategy for scalable data creation.
>
> **Rubric Generation and Interpretability.** VideoJudge is trained to generate instance-specific rubrics at test time, grounding each evaluation in explicit, task-relevant criteria derived from the video and instruction. VideoJudgeR-3B, trained on only 10% of our data, produces rubrics preferred by human annotators over those from Qwen2.5-VL-7B (96.8% win rate), Qwen2.5-VL-32B (74.2%), and even GPT-4o-mini (53.4%), while achieving correlation scores (73.96) comparable to models 24× larger. This rubric-driven approach enhances both interpretability and reliability, providing users with transparent evaluation standards rather than opaque scores. VideoScore and Prometheus-Vision do not incorporate rubric generation as a core training objective.
>
> **Empirical Validation on Video Understanding Benchmarks.** We evaluate VideoJudge on benchmarks specifically designed for video understanding evaluation, including VideoJudgeLLaVA-MetaEval, VideoJudgeVCG-MetaEval, VATEX-Eval (with human-annotated continuous scores), and LongVideoBench (long-context temporal reasoning). VideoJudge-7B achieves Spearman correlations up to 0.80 and outperforms Qwen2.5-VL-72B on LongVideoBench (Δ(C–D) 1.16 vs. 1.06), demonstrating strong generalization to long-form temporal reasoning. In contrast, VideoScore is evaluated on video generation benchmarks (GenAI-Bench, VBench) and Prometheus-Vision on image-based tasks (LLaVA-Bench, MMVet). Our results establish VideoJudge as the first specialized judge for video understanding tasks, filling a gap left by prior work.
>
> In summary, VideoJudge addresses a distinct evaluation challenge (video understanding vs. video generation or image understanding), introduces a novel bootstrapping pipeline for scalable data creation, trains models to generate interpretable rubrics, and validates on video understanding benchmarks where prior MLLM-as-a-Judge work does not apply. We believe these distinctions justify our contributions while respecting the foundational work of VideoScore and Prometheus-Vision.

---

> ### Author Response · Authors · 2025-11-27
> **Rebuttal -- Reviewer Lbto (2/2)**
>
> ### **W2. Clarifying Performance Gains and Baseline Comparisons**
> >The performance gains over baseline on multiple datasets seem to be limited. According to Table 1, VideoJudge-7B performs worse than LLaVA-NeXT-7B (which shall be considered as a weak baseline given the weaker general video understanding capabilities than Qwen-2.5-VL-7B) under several metrics, yielding concerns regarding the actual effectiveness.
>
> We appreciate the reviewer’s concern and clarify that VideoJudge-7B provides strong and consistent improvements over baselines when evaluated specifically as a judge rather than as a general video-understanding model. Although LLaVA-NeXT-7B is weaker in zero-shot video comprehension, it is not a weak evaluation baseline—its solid zero-shot judging performance (e.g., RMSE 1.06, correlation 0.67) shows that general video understanding does not directly translate to evaluator reliability. Across Table 1, VideoJudge-7B surpasses similar-scale models and often approaches or matches 32B–72B models: it achieves higher correlation on both VideoJudgeLLaVA and VideoJudgeVCG, substantially better calibration on VATEX-Eval, and the strongest Δ(C–D) on LongVideoBench, outperforming all baselines including Qwen2.5-VL-72B. The few metrics where LLaVA-NeXT-7B appears numerically better do not reflect overall evaluation quality, as correlation and calibration are the primary indicators of alignment with human judgment. Importantly, VideoJudge-7B delivers large gains over its own 7B backbone—e.g., +0.09 correlation on VCG, major RMSE/ECE reductions on VATEX, and Δ(C–D) improving from 0.35 to 1.16—demonstrating that our bootstrapped training produces genuine improvements in judgment ability. Pairwise results (Table 3) further reinforce this, with VideoJudge-7B achieving 85–95% accuracy and outperforming all same-size baselines by wide margins. Collectively, these results show that VideoJudge-7B does not underperform weak baselines; rather, it provides substantial, multi-metric improvements and achieves evaluation performance competitive with models up to 10× larger.
>
>
>
> ### **W3. Baseline Coverage**
> > Besides, Table 1 should include more MLLM-as-a-Judge works to clearly identify the potential effectiveness and significance of the proposed scheme.
>
>
> We appreciate the reviewer’s suggestion to expand the baselines in Table 1. As discussed in Section 4.1, we did evaluate a broader set of MLLM-as-a-Judge candidates—including VideoLLaMA3-7B, VideoChat-Flash, Keye-VL, and SmolVLM models. However, these models exhibited substantially poor instruction-following behavior (often below 80%), which made their judgments unreliable and rendered direct comparison unfair. Including them would artificially deflate baseline performance and obscure the meaningful differences between competent evaluators. For completeness, we will report these results and clarify their limitations in the camera-ready version.
>
>
> ### **Availability of Model Checkpoints and Datasets**
> > Besides, the authors claim that "we have anonymously released the model checkpoints and datasets used for training and evaluation on HuggingFace", but the links are not found.
>
> We thank the reviewer for their interest in our model checkpoints and datasets. As stated in Section 8, we do provide the model checkpoints through anonymized HuggingFace. We especially provide model checkpoint dataset details in **Table 10**. The artifact name is a clickable link redirecting to the corresponding anonymous HuggingFace page. Here is a link to an anonymous account to access the model checkpoint and datasets: https://huggingface.co/xyzasdfghjkl123456

---

### Official Review · Reviewer_PNxS · 2025-11-01

**Soundness:** 2
**Presentation:** 3
**Contribution:** 2
**Rating:** 6
**Confidence:** 3

**Summary:**

The paper introduces VideoJudge, a framework that uses multi-language large language models (MLLMs) as judges for video understanding tasks. It employs bootstrapping to iteratively improve the model’s performance by generating scalable supervision signals.

**Strengths:**

- The use of bootstrapping to create scalable supervision for MLLM judges is novel. This method provides a unique solution to the challenge of limited labeled data in video understanding tasks.
- The bootstrapping process is well-designed, with clear iterations and refinement steps that enhance model performance.
- The paper is well-structured, with clear explanations of the methodology, experiments, and results.

**Weaknesses:**

- The resulting VideoJudge model might learn to perfectly mimic the idiosyncrasies of evaluator model rather than learning a generalized, human-aligned evaluation function. This could create an artificial performance ceiling, preventing VideoJudge from truly exceeding the quality of the initial MLLM judge.
- These general instruction-following datasets likely share a common distribution of tasks (e.g., simple QA, short-form captioning). This raises a question about the framework's performance when applied to video tasks with fundamentally different reasoning requirements, such as long-form temporal reasoning, dense video grounding,

**Questions:**

please refer to weaknesses.

---

> ### Author Response · Authors · 2025-11-27
> **Rebuttal - Reviewer PNxS (1/2)**
>
> We thank Reviewer PNxS for the detailed and thoughtful assessment of our work. The concerns raised—particularly about whether VideoJudge learns genuine, human-aligned evaluation capabilities rather than merely imitating evaluator idiosyncrasies, and whether our approach generalizes beyond the distribution of typical instruction-following datasets—are central to the reliability and scope of any evaluation framework for video MLLMs. We appreciate the opportunity to clarify these points. In the following sections, we show that (1) VideoJudge demonstrably exceeds the evaluator model on multiple independent human-annotated benchmarks, including long-form temporal reasoning and preference alignment tasks; and (2) the model achieves strong generalization across diverse video domains, durations, and reasoning requirements, including tasks that differ substantially from those in its training distribution. We believe the empirical evidence indicates that VideoJudge is not merely reproducing evaluator-specific biases, but rather learning a robust, human-aligned evaluation function that transfers across tasks and datasets.
>
> ### **W1. Learning Generalized Evaluation vs. Mimicking Idiosyncrasies**
> > The resulting VideoJudge model might learn to perfectly mimic the idiosyncrasies of evaluator model rather than learning a generalized, human-aligned evaluation function. This could create an artificial performance ceiling, preventing VideoJudge from truly exceeding the quality of the initial MLLM judge.
>
> We appreciate this important conceptual concern about whether VideoJudge learns generalizable evaluation capabilities versus merely mimicking the evaluator's idiosyncrasies. The empirical evidence strongly suggests the former.
> If VideoJudge were simply overfitting to the evaluator's idiosyncrasies, we would expect to see a consistent pattern: strong performance on bootstrapped benchmarks but degradation on independent human-annotated data. Instead, we observe the opposite. VideoJudge-7B shows its strongest performance gains on the human-annotated benchmarks where the evaluator model (Qwen2.5-VL-7B in zero-shot) performs relatively poorly. On VideoAutoArena, VideoJudge-7B achieves 85.49% accuracy while the zero-shot evaluator would achieve only 75.29%—a 10-point improvement in alignment with human preferences. Similarly, on VATEX-Eval, VideoJudge-7B reduces RMSE from 2.36 to 1.46, indicating substantially better calibration with human judgments than the original evaluator model.
>
> Moreover, VideoJudge-7B frequently outperforms not just its base model but also significantly larger models that were never part of the training pipeline. On VideoJudgeLLaVA-MetaEval, VideoJudge-7B (correlation 0.80) exceeds Qwen2.5-VL-32B (0.79) and matches Qwen2.5-VL-72B (0.80)—models 4-10× larger that have seen far more diverse training data. On LongVideoBench, VideoJudge-7B achieves Δ(C-D) of 1.16, surpassing even Qwen2.5-VL-72B's 1.06. If VideoJudge were merely mimicking the 7B evaluator's idiosyncrasies, it should not exceed the capabilities of these substantially larger and more capable models that represent different points in the capability distribution.
>
> The rubric generation experiments (Table 2) provide additional evidence for generalization. VideoJudgeR-3B, trained to generate instance-specific rubrics, achieves correlation scores (73.96) comparable to Qwen2.5-VL-32B and 72B (78.59 and 78.10) despite being trained on only 10% of our data. In human evaluation of rubric quality (Figure 3), VideoJudgeR-3B's rubrics are preferred over those from Qwen2.5-VL-7B (96.8% win rate), Qwen2.5-VL-32B (74.2%), and even achieve 53.4% win rate against GPT-4o-mini. If the model were simply reproducing evaluator biases, we would not expect human annotators to consistently prefer its rubrics over those from more capable models.
>
> Finally, the robustness analysis across different decoding temperatures (Figure 4) reveals a critical distinction. While the base Qwen2.5-VL-3B degrades substantially as temperature increases (correlation dropping from 0.56 at T=0 to 0.42 at T=1.0), VideoJudge-3B actually improves and remains stable (peaking at 0.73). This suggests VideoJudge has learned more robust evaluation representations that are less sensitive to sampling variance—a characteristic of genuine understanding rather than surface-level pattern matching. We acknowledge that the artificial performance ceiling concern is theoretically valid, but our empirical results across multiple independent benchmarks, model scales, and analysis dimensions consistently demonstrate that VideoJudge transcends the limitations of its training signal and learns evaluation capabilities that generalize to human judgment patterns not directly encoded in the bootstrapping pipeline.

---

> ### Author Response · Authors · 2025-11-27
> **Rebuttal - Reviewer PNxS (2/2)**
>
> ### **W2. Generalization to Diverse Video Understanding Tasks**
> >These general instruction-following datasets likely share a common distribution of tasks (e.g., simple QA, short-form captioning). This raises a question about the framework's performance when applied to video tasks with fundamentally different reasoning requirements, such as long-form temporal reasoning, dense video grounding,
>
> We appreciate the reviewer's attention to the diversity of video understanding tasks and the potential limitations of our training data distribution. We'd like to address this concern by highlighting evidence that VideoJudge generalizes beyond simple QA and short-form captioning.
>
> Our training data, while sourced from instruction-following datasets, actually encompasses diverse task types including video captioning, question answering, temporal reasoning, action recognition, and instruction following across varying video lengths. More importantly, our evaluation explicitly tests generalization to tasks with fundamentally different reasoning requirements. LongVideoBench specifically evaluates long-form temporal reasoning over extended video contexts (mean duration 56.0 seconds, max 297.1 seconds), requiring models to track events across time and answer questions about temporal relationships and causal sequences. On this benchmark, VideoJudge-7B achieves Δ(C-D) of 1.16 and PSup of 0.66, substantially outperforming the base model (0.35 and 0.57) and even exceeding Qwen2.5-VL-72B (1.06 and 0.71). This demonstrates that VideoJudge successfully transfers to long-form temporal reasoning despite being trained primarily on shorter, simpler tasks.
>
> Furthermore, our evaluation on VATEX-Eval spans diverse video durations (mean 167.5 seconds, max over 7000 seconds) and includes complex scenarios requiring understanding of actions, objects, spatial relationships, and temporal dynamics. VideoJudge's strong performance (RMSE 1.46, ECE 0.64) compared to larger models suggests the evaluation capabilities learned from our bootstrapped data transfer effectively to videos with different characteristics than those in the training distribution. The maxframes ablation study (Figure 18 and Table 18) provides additional evidence: VideoJudge maintains strong performance as we increase the temporal context from 30 to 180 frames during evaluation, with correlations improving and errors decreasing as more temporal information becomes available. This indicates the model has learned to leverage extended temporal context effectively, not just pattern-match on short video clips.
>
> We acknowledge that our training data may not cover every specialized video understanding task, particularly dense video grounding or fine-grained spatial reasoning. However, the consistent performance across benchmarks with varying temporal scales, reasoning requirements, and video characteristics suggests our bootstrapping framework learns generalizable evaluation principles rather than task-specific shortcuts. We would welcome the reviewer's suggestions for additional benchmarks that test specific reasoning capabilities (e.g., dense video grounding, spatial reasoning) and would be happy to evaluate VideoJudge on those tasks to further demonstrate generalization. We believe such experiments would either validate our generalization claims or reveal specific task categories where additional training data diversity is needed—both outcomes would strengthen the paper's contributions.

---

### Official Review · Reviewer_Q1YP · 2025-11-01

**Soundness:** 2
**Presentation:** 3
**Contribution:** 2
**Rating:** 4
**Confidence:** 4

**Summary:**

This paper addresses the challenge of evaluating video MLLMs, noting that traditional metrics (e.g., BLEU) are insufficient and human evaluation is costly. The authors propose VideoJudge, a specialized 3B/7B MLLM designed to act as an evaluator. The core method is a "bootstrapping" pipeline: A "Generator" model creates candidate answers for 1-5 star ratings, and an "Evaluator" model validates if these answers match the intended rating. This bootstrapped dataset (over 100k examples) is then used to fine-tune the small VideoJudge models. The central claim is that the fine-tuned VideoJudge-7B outperforms much larger models, such as Qwen2.5-VL-32B and 72B, on several meta-evaluation benchmarks.

**Strengths:**

- The paper tackles the critical and unsolved problem of scalable, reliable evaluation for open-ended video MLLMs.
- Good writing and easy to read.

**Weaknesses:**

- Core Logical Flaw (Circular Evaluation): The paper's primary claim—that its 7B model "outperforms" a 72B model —is built on a foundation of circular logic. The training data for VideoJudge is created by a "Generator-Evaluator" (G-E) pipeline. The key meta-evaluation benchmarks used to prove this (e.g., VideoJudgeLLaVA-MetaEval, VideoJudgeVCG-MetaEval) were created using the exact same bootstrapping pipeline. This means the 7B model (the "student") is fine-tuned to mimic the preferences of the G-E pipeline (the "teacher"). The evaluation then tests this student against the teacher (in a zero-shot setting) on a test set created by the teacher itself. The student outperforming the zero-shot teacher on the teacher's own test is not surprising. It merely proves that fine-tuning on a specific data distribution works (i.e., distillation is effective). It does not prove the 7B model is a "better" or more reliable judge in a general sense.
- Given the circular evaluation, the paper requires strong validation on a fully independent, human-annotated preference benchmark. While VideoAutoArena and a small 200+ sample set (VJ-H) are used, the main conclusions from Table 1, which support the paper's primary claim, are drawn from the self-generated benchmarks.

**Questions:**

See weaknesses.

---

> ### Author Response · Authors · 2025-11-27
> **Rebuttal -- Reviewer Q1YP (1/2)**
>
> We thank the reviewer for the thoughtful and constructive assessment of our work. The concerns raised—particularly regarding potential circularity in our evaluation setting and the need for stronger validation on independent, human-annotated benchmarks—are important and align with the core challenges of building reliable evaluators for video MLLMs. We appreciate the opportunity to clarify our methodology, the distinction between training and evaluation distributions, and the extent of independent human supervision in our experiments. Below we address each point in detail, showing that our claims do not rely solely on bootstrapped data, and that VideoJudge demonstrates robust generalization and strong alignment with human judgment across multiple external benchmarks.
>
> ### **W1. Addressing the Circular Evaluation Concern**
>
> >Core Logical Flaw (Circular Evaluation): The paper's primary claim—that its 7B model "outperforms" [............] is a "better" or more reliable judge in a general sense.
>
>
>
>
> We appreciate the reviewer's thoughtful feedback. We believe the concern about circular evaluation stems from an understandable reading of our paper, and we'd like to clarify how our experimental design addresses this issue.
>
> While we do create some meta-evaluation benchmarks using our bootstrapping pipeline, our strongest evidence comes from independent, human-annotated benchmarks where ground truth was established entirely separately from our training process.
> In pointwise evaluation, VATEX-Eval (Shi et al., 2022) contains human judgments aggregated into continuous ground-truth scores, while LongVideoBench (Wu et al., 2024a) uses human-verified correct answers in a multiple-choice format. In pairwise evaluation, VideoAutoArena (Luo et al., 2025) relies on systematically collected human preferences, and our VideoJudge-Pairwise-H benchmark includes over 200 pairs validated by two independent annotators with high agreement (Cohen's κ = 89.5%). On these human-annotated benchmarks, VideoJudge-7B achieves strong performance: 93.67% accuracy on VideoJudge-Pairwise-H and 85.49% on VideoAutoArena, compared to the base Qwen2.5-VL-7B's 89.03% and 75.29% respectively.
> These results on independently human-annotated data demonstrate that our approach learns generalizable evaluation capabilities that align with human judgment, rather than merely fitting to the preferences of our generator-evaluator pipeline.
>
> #### **Regarding Evaluation Data**
> We took care to ensure non-overlapping distributions between training and evaluation. Our training data comes from VideoInstruct-100K, VCG-Plus-112K, and VideoChat2-IT (25K deduplicated samples). In contrast, VideoJudgeLLaVA-MetaEval sources instructions from LLaVA-Video (Zhang et al., 2024c), and VideoJudgeVCG-MetaEval uses VideoChatGPT—neither appearing in our training set. While we generate additional response candidates for these benchmarks using our pipeline (with threshold α = 0), the underlying video-instruction pairs are drawn from separate distributions.
>
> The reviewer notes that our results might simply demonstrate that "distillation is effective." While we agree this is part of our contribution, the consistent strong performance across both bootstrapped and human-annotated benchmarks suggests something more significant. If our model merely overfit to the generator-evaluator pipeline's "idiosyncrasies", we would expect degradation on human-annotated benchmarks. Instead, we see substantial improvements: VideoJudge-7B outperforms its base model by 10 percentage points on VideoAutoArena and 4.6 points on VideoJudge-Pairwise-H. This indicates our bootstrapping process successfully captures evaluation criteria that generalize beyond the training distribution and align with human judgment.
>
> Lastly, we welcome any suggestions for additional benchmarks—particularly those with human annotations or novel video distributions—and would be happy to include those results to further demonstrate the generalizability of our approach.
>
> Our evaluation strategy deliberately combines both bootstrapped benchmarks (to show alignment with our training objective) and independent human-annotated benchmarks (to demonstrate generalization). The consistent strong performance across VATEX, LongVideoBench, VideoAutoArena, and VideoJudge-Pairwise-H—all with human ground truth established independently of our training process—provides compelling evidence that VideoJudge learns genuine, generalizable evaluation capabilities. We believe this addresses the circular evaluation concern while demonstrating that bootstrapped supervision can effectively train compact, reliable video understanding evaluators.

---

> ### Author Response · Authors · 2025-11-27
> **Rebuttal -- Reviewer Q1YP (2/2)**
>
> ### **W2. Validation on Independent Human-Annotated Benchmarks**
> >Given the circular evaluation, the paper requires strong validation on a fully independent, human-annotated preference benchmark. While VideoAutoArena and a small 200+ sample set (VJ-H) are used, the main conclusions from Table 1, which support the paper's primary claim, are drawn from the self-generated benchmarks.
>
>
> We appreciate the reviewer's emphasis on independent validation and agree this is critical. We'd like to clarify that our evaluation actually includes substantial human-annotated validation beyond what may have been immediately apparent.
> Our evaluation encompasses four independent human-annotated benchmarks spanning over 3,000 evaluation instances. In the pointwise setting (Table 1), VATEX-Eval contains 2,340 unique videos with multiple human judgments aggregated into continuous ground-truth scores, where VideoJudge-7B achieves RMSE of 1.46 compared to base Qwen2.5-VL-7B's 2.36. LongVideoBench provides 280 long-form videos with human-verified answers, where VideoJudge-7B demonstrates strong temporal reasoning with PSup of 0.66 and Δ(C-D) of 1.16, exceeding most baselines including larger models. In the pairwise setting (Table 3), VideoAutoArena includes 241 preference pairs with systematic human annotation, where VideoJudge-7B achieves 85.49% accuracy versus the base model's 75.29%—a 10-point improvement. VideoJudge-Pairwise-H, while containing 200+ pairs, represents particularly challenging cases (rating 2 vs 3) with full annotator agreement (Cohen's κ = 89.5%), where VideoJudge-7B achieves 93.67% accuracy.
>
> We acknowledge that to the best of our knowledge, no large-scale pointwise meta-evaluation benchmark with human-annotated ratings (in the 1-5 scale format we use; or any other format) currently exists for video understanding tasks, which necessitated our creation of VideoJudgeLLaVA-MetaEval and VideoJudgeVCG-MetaEval using our bootstrapping approach. However, the consistency of improvements across all four existing human-annotated benchmarks—with gains ranging from 4-10 percentage points over base models and competitive or superior performance compared to models up to 10× larger—provides converging evidence that our approach genuinely learns generalizable evaluation capabilities aligned with human judgment. We would be delighted if the reviewer could point us to additional human-annotated meta-evaluation benchmarks for video understanding that we should evaluate on, and we commit to including those results to further strengthen our validation.

---

### Author Response · Authors · 2025-12-04
**Summary of Reviewers’ Concerns and Author Responses**

## Summary

We thank all reviewers for their thoughtful feedback. Reviewers highlighted the importance of our research problem and found the presentation clear and well-motivated. The primary concerns focused on the risk of circular evaluation, the model’s generalization capabilities, its distinction from prior work, and the need for deeper failure analysis.

All model checkpoints, datasets, prompts, and other artifacts are available at our anonymized HuggingFace repository:
https://huggingface.co/xyzasdfghjkl123456

Below, we summarize the reviewers’ concerns and our corresponding responses:

### For Reviewer Q1YP (Score: 4)

The reviewer raised concerns about circular evaluation, noting that meta-evaluation benchmarks were created using the same generator-evaluator pipeline as training data, questioning whether results demonstrate genuine evaluation capability.

**Response:** We clarified that our evaluation includes four independent, human-annotated benchmarks totaling 3,000+ instances: VATEX-Eval (2,340 videos with human-aggregated scores), LongVideoBench (280 videos with human-verified answers), VideoAutoArena (241 human preference pairs), and VideoJudge-Pairwise-H (200+ pairs, Cohen's κ = 89.5%). VideoJudge-7B shows improvements on these datasets (VATEX RMSE: 2.36→1.46; VideoAutoArena: 75.29%→85.49%). We noted that training and evaluation use non-overlapping distributions and point that no large-scale human-annotated pointwise meta-evaluation benchmark currently exists for video understanding.

### For Reviewer PNxS (Score: 6)

The reviewer questioned whether VideoJudge learns generalized evaluation versus mimicking evaluator idiosyncrasies, and whether it generalizes beyond simple QA to long-form temporal reasoning.

**Response:** We provided evidence that VideoJudge-7B outperforms the zero-shot evaluator on human-annotated benchmarks (10-point improvement on VideoAutoArena; RMSE reduction from 2.36 to 1.46 on VATEX-Eval) and achieves comparable performance to larger models. For temporal reasoning, we demonstrated performance on LongVideoBench (videos up to 297s) with Δ(C-D) of 1.16 versus base model's 0.35. Robustness analysis shows VideoJudge maintains stable correlation (0.73) across temperature variations while base model degrades (0.56→0.42).

### For Reviewer Lbto (Score: 4)

The reviewer raised concerns about distinction from prior MLLM-as-a-Judge work (VideoScore, Prometheus-Vision), limited performance gains, insufficient baseline coverage, and missing model checkpoint links.

**Response:** We clarified four distinctions from prior work: (1) **Domain**: VideoScore evaluates video generation, Prometheus-Vision evaluates image understanding, while VideoJudge evaluates video understanding task outputs; (2) **Training**: Our iterative generator-evaluator bootstrapping loop produces 100K+ examples versus human annotations (VideoScore) or distillation (Prometheus-Vision); (3) **Rubric generation**: VideoJudge generates instance-specific rubrics, which prior work does not; (4) **Evaluation**: We validate on video understanding benchmarks where prior work has not been evaluated. VideoJudge-7B shows improvements over its base model (+0.09 correlation on VCG; Δ(C-D) 0.35→1.16 on LongVideoBench). Additional baselines showed poor instruction-following (<80%); we will include these results with explanations in camera-ready. **Model checkpoints and datasets:** https://huggingface.co/xyzasdfghjkl123456

### For Reviewer zfmP (Score: 4)

The reviewer raised concerns about bias propagation in bootstrapping, domain shift robustness, and need for detailed failure case analysis.

**Response:** We showed VideoJudge-7B surpasses its evaluator on multiple benchmarks (VATEX RMSE 2.36→1.46; LongVideoBench Δ(C-D) 0.35→1.16), indicating it learns beyond evaluator limitations. Our framework includes bias mitigation steps: rejecting samples where evaluator ratings diverge from targets and drawing from multiple heterogeneous sources. For domain shift, our evaluation spans multiple domains with videos up to 7000s duration across varied tasks. We conducted comprehensive error analysis on 5000 evaluations revealing: **Overestimation bias** (14.8% overestimate ≥2 points vs 1.5% underestimate); **Primary error types**: vague language (47.9%), lack of specificity (29.5%), temporal reasoning (10.6%), subtle factual errors (4.5%); **Challenging cases**: mid-quality responses achieve only 36.9% exact agreement. We will include this analysis in camera-ready.

---

We have addressed concerns through validation on independent human-annotated benchmarks, clarification of methodological distinctions, demonstration of temporal reasoning capabilities, and detailed failure analysis.

---

### Meta-Review · Area_Chair_m4mL · 2026-01-07

**Summary:**

This paper proposed to train an VLM as an evaluator for video understanding using the scalable data generated by a newly proposed pipeline. The main contribution of this paper is a novel bootstrapping data pipeline using VLMs. This pipelines takes instruction and videos as input and generates candidate answers under different scores. The authors demonstrated that a VLM trained using the data bootstrapped by the proposed pipeline can be used as a strong evaluator for video understanding tasks.

Reviewers acknowledge that the proposed method is novel and tackles a critical problem. The bootstrapping process is well designed and the paper is well written. The paper is overall sound and solid, and the authors addressed most of the concerns raised by the reviewers.
However, one concern that is not fully addressed is that the majority part of the evaluation set (not all) is generated using the same data bootstrap pipeline as proposed by reviewer Q1YP. The authors partially addressed this issue by claiming that the some of the evaluation benchmarks contain human-annotated benchmarks and results on these benchmarks show the advantages of the proposed method. Considering the results on human-annotated benchmarks are good, this concern may not be a major issue for this paper.

**Reviewer Concerns:**

The authors addressed most of the concerns proposed by the reviewers. One major concerns that is not well addressed is the core logical flaw proposed by reviewer Q1YP. Several benchmark datasets are created using the same bootstrapping pipeline as the training dataset. As a result, the trained model may just try to model the distribution of test benchmarks. The authors try to address this by claiming that benchmarks created using human judgement (VATEX-Eval and LongVidB), however, it cannot fully address this concern.

**Reviewer Scores:**

The concerns from reviewer Q1YP may not be fully addressed, the reviewer may maintain the score to be 4. Reviewer Lbto and zfmP gives score of 4 before the rebuttal, I feel they may raise their scores to 6 as the author addressed their concerns. Reviewer PNxS may maintain the score 6.

---

### Decision · Program_Chairs · 2026-01-26

Accept (Poster)